# Towards Calibrated Robust Fine-Tuning of Vision-Language Models

**Changdae Oh**[*,c,n]
University of Wisconsin–Madison

**Hyesu Lim**[*,c]
KAIST AI

**Mijoo Kim**[c]
Chung-Ang University

**Dongyoon Han**
NAVER AI Lab

**Sangdoo Yun**
NAVER AI Lab

**Jaegul Choo**
KAIST AI

**Alexander Hauptmann**
Carnegie Mellon University

**Zhi-Qi Cheng**†
Carnegie Mellon University

**Kyungwoo Song**†
Yonsei University

## Abstract

Improving out-of-distribution (OOD) generalization during in-distribution (ID) adaptation is a primary goal of robust fine-tuning of zero-shot models beyond naive fine-tuning. However, despite decent OOD generalization performance from recent robust fine-tuning methods, confidence calibration for reliable model output has not been fully addressed. This work proposes a robust fine-tuning method that improves both OOD accuracy and confidence calibration simultaneously in vision language models. Firstly, we show that both OOD classification and OOD calibration errors have a shared upper bound consisting of two terms of ID data: 1) ID calibration error and 2) the smallest singular value of the ID input covariance matrix. Based on this insight, we design a novel framework that conducts fine-tuning with a constrained multimodal contrastive loss enforcing a larger smallest singular value, which is further guided by the self-distillation of a moving-averaged model to achieve calibrated prediction as well. Starting from empirical evidence supporting our theoretical statements, we provide extensive experimental results on ImageNet distribution shift benchmarks that demonstrate the effectiveness of our theorem and its practical implementation. Our code is available here.

## 1 Introduction

Foundation models [6] such as CLIP [47] have been extensively utilized on diverse domains via pretrain-finetune approaches. Their generalized knowledge shaped after large-scale pre-training enables them to easily adapt to downstream tasks through zero-shot inference or fine-tuning. However, it has been steadily reported that a naive fine-tuning approach comprises foundation models' strong out-of-distribution (OOD) generalization capability during adaptation to in-distribution (ID) data [61, 30]. To ensure robustness under distribution shifts, a wide range of research has followed [61, 30, 17, 32, 57, 43, 42] so-called *robust fine-tuning*. Despite the advancements of the robust fine-tuning methods, we are aware that an important criterion for trustworthy machine learning has been overlooked – *confidence calibration* [39, 8] that quantifies how close the confidence of our predictor is to the actual correctness of predictions. As shown in Figure 1, existing robust fine-tuning methods hurt the confidence calibration in terms of expected calibration error (ECE) [40] on OOD data compared to the zero-shot evaluation while they show improvements on OOD accuracy. In this

---

[*]Equal contribution (changdae@cs.wisc.edu; hyesulim@kaist.ac.kr), Work partly done at [c]Carnegie Mellon University and [n]NAVER AI Lab, †Co-correspondence (zhiqic@cs.cmu.edu; kyungwoo.song@yonsei.ac.kr)

38th Conference on Neural Information Processing Systems (NeurIPS 2024).

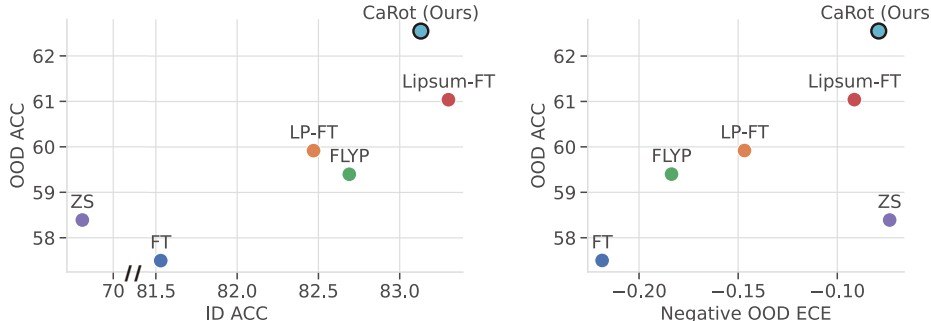

Figure 1: **OOD accuracy vs. ID accuracy (left) and negative OOD ECE (right).** To maintain consistency in the plots, where desired values are shown on the right side of the x-axis, we report negative OOD ECE. ID ACC refers to ImageNet-1K top-1 accuracy; OOD ACC and ECE refer to the averaged accuracy and ECE of the five ImageNet distribution shifts (ImageNetV2, ImageNet-R, ImageNet-A, ImageNet-Sketch, and ObjectNet), respectively. Detailed numbers are reported in Table 2 and 3. Note that the competing methods – FLYP [17], LP-FT [30], and Lipsum-FT [42] – improve OOD accuracy over the zero-shot baseline (ZS) and naive fine-tuning (FT) but suffer from OOD miscalibration, presumably due to concerning generalization solely during fine-tuning. Our CaRot outperforms existing methods on both OOD accuracy and calibration by large margins.

work, we introduce a calibrated robust fine-tuning method (**CaRot**) that simultaneously improves confidence calibration and accuracy of the classifier on OOD data.

Confidence calibration is a key aspect of reliable machine learning, essential for avoiding high-confidence incorrect predictions in real-world decision-making systems. This is particularly crucial in high-stakes tasks like autonomous driving and healthcare applications. After a seminal work [18] revealed the miscalibration problem of high-performing neural networks, a plethora of attempts followed to improve the calibration of neural network models through post-hoc adjustments [65, 18, 29, 68, 19] or train-time regularizations [67, 52, 56, 38, 37]. However, many of them focus on improving calibration for ID samples, and methods for enhancing OOD calibration usually require OOD samples at train time [63, 16]. Moreover, these approaches commonly focus on calibration alone without ensuring improvement in other quantities, e.g., accuracy. In this work, we explore a unified framework that jointly considers calibration and accuracy (particularly on OOD data).

To accomplish low classification and calibration errors on OOD samples with only ID samples in our hands, we conduct theoretical analyses of those OOD errors. To be specific, we derive an upper bound that is shared for OOD classification error and OOD calibration error composed with two quantities on ID samples, 1) *the reciprocal of the smallest singular value of the normalized covariance matrix of ID data representation* and 2) *the ID calibration error*. Different from the existing bounds focusing on either one of classification or calibration error [5, 72, 63], we address both classification and calibration errors in a single unified bound. More importantly, the chief components of our bound can be computed solely with ID samples without relying on any OOD samples, which discerns our approach to existing work [4].

Motivated by our theoretical analysis, we propose **a new multimodal contrastive loss that promotes the smallest singular value of input image representation to become larger** by enforcing the orthogonality of the final projection matrix of the visual encoder. Furthermore, to understand the working mechanism in depth, we present an interpretation of our new multimodal contrastive loss as a process of seeking the low-rank approximation of cross-covariance matrix over image-text representations on a reduced solution space induced by the orthogonality constraint. Meanwhile, to **enhance confidence calibration on ID samples during fine-tuning**, we utilize a self-distillation (SD) with an exponential moving average (EMA) teacher model. This EMA SD encourages a student model to learn semantic similarity structures of in-batch data representations from teacher predictions across diverse contrastive pairs, appropriately adjusting confidence per instance.

We first validate our new error bounds with synthetic datasets to show that the bounds hold empirically. Then, evaluate our method by conducting extensive experiments of fine-tuning CLIP [47] on ImageNet-1K [10] classification task under natural distribution shift (ImageNet-V2/R/A/Sketch and ObjectNet) and synthetic distribution shift (ImageNet-C). We demonstrate the effectiveness of our proposed framework for robust generalization and calibration by observing consistent improvements in terms of expected calibration error and accuracy on ID/OOD datasets.

**Summary of contributions.** 1) We point out that existing fine-tuning methods do not adequately achieve satisfactory OOD generalization and calibration simultaneously. 2) We provide theoretical analysis for classification and calibration errors on the OOD data and show that they are both bounded from above by the ID calibration error and the smallest singular value of the covariance matrix over the ID input representation. 3) Based on our theoretical analyses, we devise a calibrated robust fine-tuning method, CaRot, as a practical realization of our theorem that reduces the upper bound of OOD classification and calibration errors by conducting constrained multimodal contrastive learning with EMA self-distillation. 4) We present empirical evidence for our theory on a synthetic dataset and demonstrate the efficacy of CaRot via extensive evaluations on ImageNet-1K distribution shifts in terms of accuracy and calibration error on ID and OOD domains.

## 2 Preliminary

**Robust fine-tuning** aims to achieve consistently high performance on data from both training distribution (ID) and related but different test distributions (OOD). For validation, we commonly consider a covariate shift scenario for the classification task, where both ID and OOD domains share the class categories ($\mathcal{Y}_{\text{ID}} = \mathcal{Y}_{\text{OOD}}$) and have the same conditional distribution $P(Y|X)$, but have different marginal distributions over input $X$. That is, $P_{\text{ID}}(Y|X) = P_{\text{OOD}}(Y|X)$ but $P_{\text{ID}}(X) \neq P_{\text{OOD}}(X)$. Here, we evaluate a model that is fine-tuned on a training split of the ID domain, on a test split of ID, and on OOD domains. The term "OOD" is quite general, and we confine the scope of OOD to transformed and related distributions with ID [12]. For example, if our ID data is about an object recognition task with images, OOD data is about a sensor-noised version of ID, or independently collected data from a different domain targeting the same task.

**Confidence calibration** is a concept of matching the prediction probabilities yielded for different inputs to the expected accuracy on these inputs. In a $K$-way classification setting, let $X \in \mathbb{R}^d$ and $Y \in \{1, ..., K\}$ be random variables indicating inputs and labels, respectively. A dataset with $N$ independent samples from the joint distribution $P(X, Y) = P(Y|X)P(X)$ is denoted by $\{(x_n, y_n)\}_{n=1}^N$. Let $f$ be a classifier and $f(y|x) = \hat{p}$ be a confidence, i.e., the maximum of probabilities among $K$ dimensions corresponding to its prediction $\hat{y}$. We say a model is *perfectly-calibrated* when $\mathbb{P}(\hat{y} = y|\hat{p} = p) = p$, $\forall p \in [0, 1]$. As a quantitative measure, the model calibration can be derived as $\mathbb{E}[|\mathbb{P}(\hat{y} = y|\hat{p} = p) - p|]$. In practice, we use expected calibration error (ECE) [40] as an empirical approximation of the model calibration, which is a weighted average of bin-wise miscalibration. The ECE divides the confidence score of $N$ samples into $M$ uniform confidence bins $\{B_m\}_{m=1}^M$ and takes the mean of the gap between accuracy (acc) and confidence (conf) over the bins weighted by the number of samples in the bins, i.e., $\text{ECE} = \sum_{m=1}^M \frac{|B_m|}{N} |\text{acc}(B_m) - \text{conf}(B_m)|$.

## 3 Theoretical Analysis on OOD Generalization and Calibration

We first identify the factors that affect OOD generalization and calibration errors under circumstances where only ID data is accessible. We take inspiration from the generalization bound of domain adaptation literature [5, 72] while remarkably adapting the analysis to consider both OOD classification error and OOD calibration error at the same time in a more practical way.

Let $\mathcal{D}$ be a domain on input space $\mathcal{X}$ and $\mathcal{Y} = \{0, 1\}$ be a label space for a binary classification. Among the sufficiently expressive hypothesis functions $h : \mathcal{X} \to [0, 1]$ in a class $\mathcal{H}$, we define $h_0(\cdot)$ as a desired calibrated predictor for $y$, which minimizes the calibration error $\mathbb{E}_{x \sim \mathcal{D}}[(h(x) - c(x))^2]$ [39], where $c(x) = \mathbb{E}_y[y|h(x)]$ is the expected value of $y$ given a prediction $h(x)$. That is, $h_0$ always produces the calibrated prediction for $y$ given $x$ so that the output confidence $h_0(x)$ matches the expectation of $y$ over the subset of samples that have the same confidence value with $h_0(x)$. Our goal is to learn a hypothesis function $h(\cdot)$ that outputs reliable prediction probability on samples from the unseen OOD domain, which is defined by a distribution $\mathcal{D}_{\text{OOD}}$, as well as on the ID domain $\mathcal{D}_{\text{ID}}$, where the predictor is trained on. In essence, the error $\varepsilon_{\mathcal{D}}(h) = \mathbb{E}_{x \sim \mathcal{D}}[(h(x) - h_0(x))^2]$ should be small for two different domains $\mathcal{D} \in \{\mathcal{D}_{\text{ID}}, \mathcal{D}_{\text{OOD}}\}$. Here, we focus on the covariate shift scenario (§2) that only the marginal distribution over $\mathcal{X}$ changes while the distribution over $\mathcal{Y}$ is preserved.

Let the optimal hypothesis $h^*$, which minimizes a combination of errors on both ID and OOD, be $h^* := \arg\min_{h \in \mathcal{H}} \varepsilon_{\mathcal{D}_{\text{ID}}}(h) + \varepsilon_{\mathcal{D}_{\text{OOD}}}(h)$, and $\Delta$ denote the optimal joint error $\Delta := \varepsilon_{\mathcal{D}_{\text{ID}}}(h^*) + \varepsilon_{\mathcal{D}_{\text{OOD}}}(h^*)$. Now, we derive a new bound for the OOD calibration error and OOD classification error.

**Theorem 3.1.** *Let $h : \mathcal{X} \to [0,1]$ be a real-valued function of structure $h(x) = \sum_{i=1}^{d} h_i(x[i])$ where $h_i$ is an arbitrary one-dimensional function, and $h$ is in a hypothesis class $\mathcal{H}$ that has pseudo dimension $\mathcal{P}dim(\mathcal{H}) = d_h$, $\hat{\mathcal{D}}_{ID}$ be an $N$-size empirical distribution on ID domain. If $(x[1], ..., x[d])$ have matching marginals for ID and OOD, and $(x[i], x[j])$ is a bi-variate Gaussian for every $i, j \in [d]$, then for any $\delta \in (0,1)$ and for all $h$, the following bounds hold with probability at least $1 - \delta$:*

$$i) \quad \varepsilon_{\mathcal{D}_{OOD}}(h) \leq \varepsilon_{\hat{\mathcal{D}}_{ID}}(h) + \frac{d}{\sigma_{min}(\tilde{\Sigma}_{\mathcal{D}_{ID}})} + \Delta + \mathcal{O}\left(\sqrt{\frac{1}{N}\log{(\frac{N}{d_h})}^{d_h}(\frac{1}{\delta})}\right) \tag{1}$$

$$ii) \quad \mathbb{E}_{\mathcal{D}_{OOD}}[(h(x) - y)^2] + \mathbb{E}_{\mathcal{D}_{OOD}}[c(x)^2] - 1 \leq \varepsilon_{\hat{\mathcal{D}}_{ID}}(h) + \frac{d}{\sigma_{min}(\tilde{\Sigma}_{\mathcal{D}_{ID}})} + \Delta + \mathcal{O}\left(\sqrt{\frac{1}{N}\log{(\frac{N}{d_h})}^{d_h}(\frac{1}{\delta})}\right) \tag{2}$$

where $\tilde{\Sigma}_{\mathcal{D}_{ID}} := \mathbb{E}_{\mathcal{D}_{ID}}[\tilde{x}\tilde{x}^T]$ is a covariance matrix with a strictly positive minimum singular value of $d$-dimensional normalized input $\tilde{x} = (\tilde{x}[1], ..., \tilde{x}[d])$, where $\tilde{x}[i] := (x[i] - \mathbb{E}[x[i]])\text{Var}(x[i])^{-1/2}$ and $\sigma_{\min}(M)$ is the smallest singular value of a matrix $M \in \mathbb{R}^{d_1 \times d_2}$. These theoretical results can be directly applied to the intermediate or penultimate layer's representation of a neural network by setting the input variable $x$ as a representation vector as in [4, 73]. From now on, we will assume the input as an image representation from the last layer of the visual encoder in the following sections. Note that 1) the LHS of the first inequality (ineq.(11)) indicates **OOD calibration error**, 2) two terms in the LHS of the second inequality (ineq.(12)) denote **OOD classification error** in terms of $L_2$ loss and prediction sharpness on OOD domain, and 3) both inequalities have the same RHS, which contains the empirical estimate of **ID calibration error**, the reciprocal of the **smallest singular value of ID input covariance matrix**, and remaining irreducible terms that depend on the problem setup. Intuitively, Theorem 3 implies that *pursuing diverse input features while maintaining the calibration of the classifier contributes to improving OOD calibration and generalization simultaneously*. We defer the proof and discussion on the tightness of the bound and its assumptions in Appendix C.

Based on our analysis, we expect the potential of reducing the upper bound of OOD calibration error and the sum of OOD classification error and prediction sharpness by minimizing the first two terms of RHS in both bounds: the empirical ID calibration error and the reciprocal of minimum singular value of the normalized ID covariance matrix. In §4, we devise a realization of this theoretical concept.

## 4 Method

Our goal is to achieve good OOD generalization and calibration during the ID adaptation of pre-trained models. Motivated by Theorem 3, we propose a new fine-tuning method that increases the smallest singular value of the ID input covariance matrix while improving ID calibration, thereby lowering the upper bound of OOD calibration and generalization errors. By following [61, 30, 17], we set a vision-language model (VLM), CLIP [47] as our target, which serves as a remarkably strong backbone for zero-shot inference and fine-tuning with ease. We limit the scope of validation to image classification tasks. Note that our theorem is not confined to specific domains or model architectures and thus can be applied beyond CLIP's image classification. See Figure 2 for an overview.

### 4.1 Robust fine-tuning with constrained multimodal contrastive learning

To adapt a pre-trained VLM on image classification tasks, the cross-entropy loss is the most common choice as an objective function. However, there are emerging shreds of evidence supporting the use of contrastive loss (CL) for robust adaptation [70, 50, 17], especially when the model is pre-trained via CL. Witnessing its empirical success on OOD generalization [17], we leverage a CL-based learning strategy for VLM fine-tuning. CLIP consists of an image encoder $f_{\theta_v}(\cdot) = f_{\hat{\theta}_v}(\cdot)W_v$ and a text encoder $g_{\theta_l}(\cdot) = g_{\hat{\theta}_l}(\cdot)W_l$, where encoders are composed with backbone models $(f_{\hat{\theta}_v}(\cdot), g_{\hat{\theta}_l}(\cdot))$ and projection matrices $(W_v \in \mathbb{R}^{d_v \times r}, W_l \in \mathbb{R}^{d_l \times r})$. The encoders produce $L_2$-normalized representations to compute the similarity between image and text inputs. Given $N$ pairs of (image, text) $\{(I_i, T_i)\}_{i=1}^{N}$, a common form of multimodal contrastive loss (MCL) can be written as

$$\mathcal{L}_{\text{MCL}}(\theta) := \frac{1}{2N}\sum_{i=1}^{N} -\log\frac{\exp(f_{\theta_v}(I_i) \cdot g_{\theta_l}(T_i))}{\sum_{j=1}^{N}\exp(f_{\theta_v}(I_i) \cdot g_{\theta_l}(T_j))}$$
$$+ \frac{1}{2N}\sum_{i=1}^{N} -\log\frac{\exp(f_{\theta_v}(I_i) \cdot g_{\theta_l}(T_i))}{\sum_{j=1}^{N}\exp(f_{\theta_v}(I_j) \cdot g_{\theta_l}(T_i))} + R(\theta_v, \theta_l), \tag{3}$$

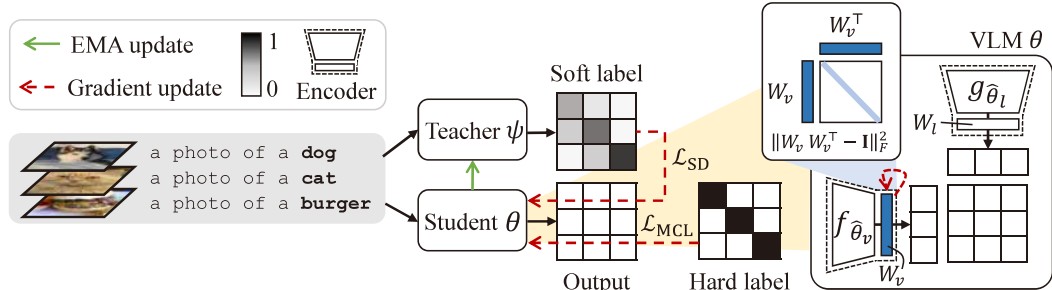

Figure 2: **Overview of CaRot**. We fine-tune a VLM using a multimodal contrastive loss with an orthogonality constraint on visual projection layer (eq.(4)) and self-distillation $\mathcal{L}_{\text{SD}}$ (eq.(5)) that takes predictions of EMA teacher $\psi$ as soft target labels to train the student model $\theta$. The darker and the lighter elements denote values closer to 1 and 0, respectively. Both teacher and student models share identical VLM architecture consisting of image $f_{\theta_v} := [f_{\hat{\theta}_v}; W_v]$ and text $g_{\theta_l} := [g_{\hat{\theta}_l}; W_l]$ encoders, where $W$ is the last projection layer. Given (image, text) pair data, the model outputs the pair-wise similarity score for in-batch image-text representations.

where $\theta = \{\theta_v, \theta_l\}$ are the parameters of image and text encoders and $R(\theta_v, \theta_l)$ reflects a general regularization strategy in CL [7, 21]. We update both image and text encoders during fine-tuning as done in the pre-train phase and use OpenAI templates [47] to create (image, text) pairs from a downstream classification dataset that consists of (image, class) pairs.

Meanwhile, the basic form of $\mathcal{L}_{\text{MCL}}$ does not inform anything about the singular value distribution of learned representation. In §3, we showed that the reciprocal of the smallest singular value constitutes the shared upper bound, i.e., the larger the smallest singular value is, the lower the upper bound becomes. To encourage this, we put a soft constraint term to $\mathcal{L}_{\text{MCL}}$ that enforces the final projection matrix $W_v$ of the visual encoder to be orthogonal and hence the output image representation matrix to have a large effective rank, as in below:

$$\mathcal{L}_{\text{MCL-con}}(\theta) := \mathcal{L}_{\text{MCL}}(\theta) + \lambda_{\text{OC}}\mathcal{L}_{\text{OC}}(W_v), \quad \mathcal{L}_{\text{OC}}(W_v) = ||W_v^T W_v - \mathbf{I}||_F^2, \tag{4}$$

where $\mathbf{I}$ is an identity matrix that has the same shape with $W_v^T W_v$ and $\lambda_{\text{OC}}$ is a strength of the orthogonality constraint[1]. While recklessly increasing the singular values might hinder ID adaptation, our orthogonal constraint mitigates the degradation of performance by pursuing not only the smallest singular values to be large but also the largest singular values to be small which is important for generalization on ID data [69]. Interestingly, this contrastive loss with regularization terms can be viewed as a constrained singular value decomposition (SVD) with a cross-covariance matrix of image-text representations where the orthogonality constraint is applied.

To be specific, by following Nakada et al. [41], under a linear representation assumption, a gradient descent step of $\hat{\mathcal{L}}_{\text{MCL-con}}$ boils down to the maximization of the SVD objective, which aims to find a low-rank approximation of the normalized cross-covariance matrix $S(\beta)$[2] as follow:

$$\arg\min_{W_v, W_l} \mathcal{L}_{\text{MCL-con}}(W) := \frac{1}{2N}\sum_{i=1}^{N} -\log \frac{\exp(W_v\hat{I}_i \cdot W_l\hat{T}_i)}{\sum_{j=1}^{N}\exp(W_v\hat{I}_i \cdot W_l\hat{T}_j)}$$

$$+ \frac{1}{2N}\sum_{i=1}^{N} -\log \frac{\exp(W_v\hat{I}_i \cdot W_l\hat{T}_i)}{\sum_{j=1}^{N}\exp(W_v\hat{I}_j \cdot W_l\hat{T}_i)} + R(W_v, W_l) + \lambda_{\text{OC}}||W_v^T W_v - \mathbf{I}||_F^2$$

$$\approx \arg\max_{W_v, W_l} \text{SVD}(S(\beta)) := \text{tr}(W_v^T S(\beta)W_l) - (\rho/2)||W_vW_l^T||_F^2 \quad \text{subject to } ||W_v^T W_v - \mathbf{I}||_F^2 = 0,$$

where we adopt $R(W_v, W_l) = (\rho/2)||W_vW_l^T||_F^2$ for $\rho > 0$ as a regularization term to promote the encoders to capture diverse features as in Ji et al. [27]. Here, we assume that the input of $\mathcal{L}_{\text{MCL-con}}$ is the penultimate representation of VLM's encoders, i.e., $(\hat{I}_i, \hat{T}_i) = (f_{\hat{\theta}_v}(I_i), g_{\hat{\theta}_l}(T_i))$, and

---

[1]While it is also possible to inject constraint on the text projection matrix $W_l$, we only do it for $W_v$ because our concern is about the singular value of image representations for downstream tasks. We observed degradations of accuracy (ID:$-0.01$, OOD:$-0.15$) and ECE (ID:$-0.002$, OOD:$-0.02$) by adding the constraint on $W_l$.

[2]$S(\beta) := \frac{1}{N}\sum_{i=1}^{N}\beta_i\hat{I}_i\hat{T}_i^T - \frac{1}{N}\sum_{i\neq j}\beta_{ij}\hat{I}_i\hat{T}_j^T$, where $\beta_i$ and $\beta_{ij}$ depend the choice of non-linear function over dot product between image and text representations. See Nakada et al. [41] for a detailed derivation

$W = \{W_v, W_l\}$ is a set of projection matrices (i.e., last layers of each encoder). This connection between $\mathcal{L}_{\text{MCL}}$ and SVD allows us to understand the working mechanism of our proposed objective. That is, minimizing $\mathcal{L}_{\text{MCL-con}}$ can be interpreted as finding a good rank-$r$ (dimensionality of the image-text projection space) approximation of the cross-modal covariance matrix by seeking the direction of large co-variation among image-text representations, while the solution space is constrained by enforcing an orthogonality condition on the collection of vision-side singular vectors $W_v$ to achieve the larger effective rank of both the projection matrix $W_v$ and the image representation matrix (See Appendix §B for further explanation on the effective rank and the smallest singular value). In §5, we validate that $\mathcal{L}_{\text{MCL-con}}$ significantly increases the smallest singular values and results in better OOD generalization and calibration on downstream tasks.

### 4.2 Calibration during robust fine-tuning

In the previous section, we devise a new multimodal contrastive loss that promotes large $\sigma_{min}(\tilde{\Sigma}_{\mathcal{D}_{\text{ID}}})$. We now address the next component standing for ID calibration, which is another crucial component according to our theoretical analysis. While there are numerous approaches to enhance calibration during neural network training [38, 56, 52, 1], we notice the promising results of knowledge distillation-based calibration approaches [64, 71]. These approaches encourage the model to learn from input-dependent smoothed labels that effectively mitigate the overconfidence issue, which is commonly associated with miscalibration. Therefore, we employ a self-distillation (SD) method for ID calibration in that distilling the similarity score map would help avoid overconfidence.

Specifically, we first initialize both teacher and student networks with a pre-trained CLIP model (including both image and text encoders); update the student model using gradient descent for every iteration while slowly updating the teacher model that has $\psi = \{\psi_v, \psi_l\}$ as parameters using EMA with the momentum of $\alpha$ at every $t > 1$ iteration, i.e., $\psi \leftarrow \alpha\psi + (1-\alpha)\theta$. Rather than hosting another VLM or fixed pre-trained CLIP as a teacher model, we adopt a self-evolving EMA network as a teacher, observing its successful usage on the weight-space ensemble between homogeneous models [61, 48], robust self-supervised learning methods [2, 45], as well as regularization [71]. With the EMA teacher $\{f_{\psi_v}(\cdot), g_{\psi_l}(\cdot)\}$ and the learning student $\{f_{\theta_v}(\cdot), g_{\theta_l}(\cdot)\}$, we construct a self-distillation loss term for $N$ data pairs as:

$$\mathcal{L}_{\text{SD}}(\theta) := \frac{1}{N}\sum_{i=1}^{N}[KL(\tilde{q}_i^I||q_i^I) + KL(\tilde{q}_i^T||q_i^T)], \tag{5}$$

where $KL$ denotes Kullback–Leibler divergence, $q_i^I = \texttt{softmax}(\{f_{\theta_v}(I_i) \cdot g_{\theta_l}(T_j)\}_{j=1}^N)$ and $q_i^T = \texttt{softmax}(\{f_{\theta_v}(I_j) \cdot g_{\theta_l}(T_i)\}_{j=1}^N)$ are student outputs, and $\tilde{q}_j^I$ and $\tilde{q}_j^T$ are teacher outputs which are similarly defined by replacing the student parameter $\theta$ with that of teacher's $\psi$. Presumably, label smoothing (LS) [55] behaves similarly to what we intended, but we argue that LS would be less effective than EMA SD in terms of mitigating overconfidence issues. See Appendix §B.

We complete the learning objective as a summation of $\mathcal{L}_{\text{MCL-con}}$ and $\mathcal{L}_{\text{SD}}$ with a coefficient $\lambda_{\text{SD}}$, i.e., $\mathcal{L} = \mathcal{L}_{\text{MCL}} + \lambda_{\text{OC}}\mathcal{L}_{\text{OC}} + \lambda_{\text{SD}}\mathcal{L}_{\text{SD}}$. The novel combination of these two components contributes to ensuring a larger smallest singular value of image representation and ID calibration simultaneously, which induces calibrated robust fine-tuning (**CaRot**) on distribution shifts. Note that this objective function is just one of the possible realizations of our upper bound (Theorem 3) on OOD generalization and calibration errors. Further exploration can spawn a more practical algorithm in the future.

## 5 Experiments

In § 5.1, we first show empirical evidence of the error bounds that we derived in §3. We then provide experimental setup and main benchmarking results (§5.2) and present further empirical studies (§5.3).

### 5.1 Numerical analysis on error bounds

Our theoretical analysis in §3 revealed the possibility of managing OOD classification and calibration errors simultaneously by leveraging a shared quantity over the ID domain (sum of calibration error term and the singular value term). Before conducting real-world evaluations, we verify the theoretical analysis with a toy experiment that simulates distribution shifts. To be specific, we generate a binary classification dataset with 1000-dimensional Gaussian random variables as features where the mean of features are partly shifted across different test environments (ID, OOD). We train a

three-layer network with regularization terms: Born-Again-Network (BAN)-style self-distillation [13] for calibration ($\mathcal{L}_{SD}$) and the orthogonal constraint (shown in eq.(4)) for the singular value ($\mathcal{L}_{OC}$). Detailed descriptions of the experimental setup are provided in Appendix §A.1.

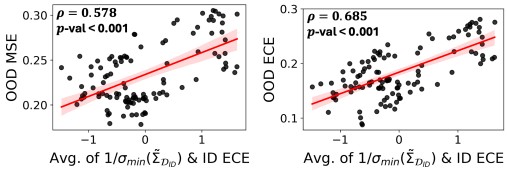

Figure 3: **Analysis of error bounds on synthetic data.** Plots on the left side show RHS (x-axis) and LHS (y-axis; MSE for ineq.(12) and ECE for ineq.(11)) of the inequalities in §3. We denote MSE for the mean squared error, $\mathcal{L}_{OC}$ for the singular value regularization, and $\mathcal{L}_{SD}$ for the calibration regularization.

Table 1: The best case values of two terms of RHS (ID $\sigma_{min}$ and ID ECE) and LHS – OOD errors (MSE and ECE) in the bounds of Theorem 3. Reported values are an average of three repeated runs.

| Method | ID | | OOD | |
|---|---|---|---|---|
| | $\sigma_{min}$ ($\uparrow$) | ECE ($\downarrow$) | MSE ($\downarrow$) | ECE ($\downarrow$) |
| Baseline | 2.0887 | 0.1666 | 0.2581 | 0.2477 |
| $\mathcal{L}_{OC}$ | 4.9630 | 0.1528 | 0.1932 | 0.1781 |
| $\mathcal{L}_{SD}$ | 3.1354 | 0.1308 | 0.2170 | 0.1720 |
| $\mathcal{L}_{OC}, \mathcal{L}_{SD}$ | 6.5961 | 0.1391 | 0.1877 | 0.1596 |

Figure 3 visualizes the results of Pearson correlation analysis between the average of $1/\sigma_{min}(\tilde{\Sigma}_{\mathcal{D}_{ID}})$ and ECE from ID samples and OOD MSE/ECE over 111 trained models. Here, we observe strong correlations between the average of $1/\sigma_{min}(\tilde{\Sigma}_{\mathcal{D}_{ID}})$ and ID ECE (x-axis), and OOD classification and calibration errors (y-axis). Additional results on the best models per each regularization term are showcased on the Table 1, which also indicates that reducing the upper bound results in better OOD generalization and calibration. These analyses demonstrate that Theorem 3 empirically holds.

## 5.2 Evaluation on distribution shift benchmarks

**Training and evaluation.** We adopt CLIP ViT-B/16 as our VLM backbone and evaluate each fine-tuning method, including CaRot, in terms of calibration (with ECE) and accuracy under distribution shifts. For downstream tasks, we consider the ImageNet-1K (IN) classification and regard it as our ID domain. For all methods, we optimize the model parameters using the AdamW with a batch size of 512 over 10 epochs. Fine-tuned models are evaluated under varying distribution shifts.

**Benchmark datasets.** We consider IN-V2 [49], IN-R [23], IN-A [24], IN-S [60], and ObjectNet [3] as natural shifts of the in-distribution dataset (IN). We refer to the average performance over these five datasets as Avg. Shifts or OOD throughout the following sections unless it is specified as a different dataset, e.g., IN-C [22] which we adopt as a synthetic shift scenario occurred by sensory noises.

Table 2: **ImageNet accuracy**. We report the accuracy on ImageNet and its distribution shift variants by fine-tuning CLIP ViT-B/16 with five methods. The best and the second-best in each column are underlined.

| Method | IN$\uparrow$ | IN-V2$\uparrow$ | IN-R$\uparrow$ | IN-A$\uparrow$ | IN-S$\uparrow$ | ObjectNet$\uparrow$ | Avg. shifts$\uparrow$ |
|---|---|---|---|---|---|---|---|
| ZS | 68.33 | 61.93 | 77.71 | 49.95 | 48.26 | 54.17 | 58.39 |
| FT | 81.53 | 71.66 | 70.14 | 44.01 | 49.11 | 52.56 | 57.50 |
| LP-FT | 82.47 | 72.71 | 72.84 | 49.31 | 50.28 | 54.45 | 59.92 |
| FLYP | 82.69 | 72.73 | 71.35 | 48.52 | 49.84 | 54.86 | 59.40 |
| Lipsum-FT | 83.30 | 73.60 | 75.90 | 49.90 | 51.40 | 54.38 | 61.04 |
| CaRot (Ours) | 83.13 | 74.11 | 77.71 | 51.60 | 52.71 | 56.60 | 62.55 |

Table 3: **ImageNet ECE.** Along with Table 2, we report the ECE on ImageNet and its distribution shifts to compare with other fine-tuning methods, which demonstrates our out-of-distribution (OOD) calibration performance. The best and the second-best in each column are underlined (See Figure B for details).

| Method | IN$\downarrow$ | IN-V2$\downarrow$ | IN-R$\downarrow$ | IN-A$\downarrow$ | IN-S$\downarrow$ | ObjectNet$\downarrow$ | Avg. shifts$\downarrow$ |
|---|---|---|---|---|---|---|---|
| ZS | 0.0570 | 0.0548 | 0.0541 | 0.0967 | 0.0850 | 0.0780 | 0.0736 |
| FT | 0.0884 | 0.1468 | 0.1164 | 0.3000 | 0.2544 | 0.2753 | 0.2186 |
| LP-FT | 0.0505 | 0.0894 | 0.0613 | 0.2051 | 0.1659 | 0.2124 | 0.1468 |
| FLYP | 0.0635 | 0.1171 | 0.0967 | 0.2435 | 0.2200 | 0.2383 | 0.1836 |
| Lipsum-FT | 0.0384 | 0.0516 | 0.0426 | 0.1290 | 0.1023 | 0.1315 | 0.0914 |
| CaRot (Ours) | 0.0470 | 0.0367 | 0.0575 | 0.1240 | 0.0699 | 0.1075 | 0.0791 |

**Baseline methods.** We benchmark CaRot alongside zero-shot inference (ZS) and fine-tuning methods: standard fine-tuning (FT), LP-FT [30], FLYP [17], and Lipsum-FT [42]. Refer §A for further details. In Appendix Table B, C, and D, we compare results with post-hoc robustification method (weight ensemble; WiSE-FT [61]) and post-hoc calibration (temperature scaling; TS [18]) method.

**Results on natural shifts.** Table 2 and 3 highlight our argument that CaRot significantly enhances both generalization and calibration on OOD data. Under distribution shifts from IN to -V2, -R, -A, -S, and ObjectNet, CaRot favorably compares with the existing best fine-tuning methods by margin of 1.51 and 0.0123 for OOD top-1 accuracy and ECE, respectively, averaged over five shifted datasets. See reliability diagrams in Appendix Figure B for deeper insight on calibration. We further report the results with different backbone models, RN50 and ViT-L/14, in Table 7 (See Table H and I for details). CaRot consistently outperforms the baseline methods for these backbones, too. Furthermore, in Table 6, we provide additional comparisons with CAR-FT [34], Model Stock [26] and ARF [20][3].

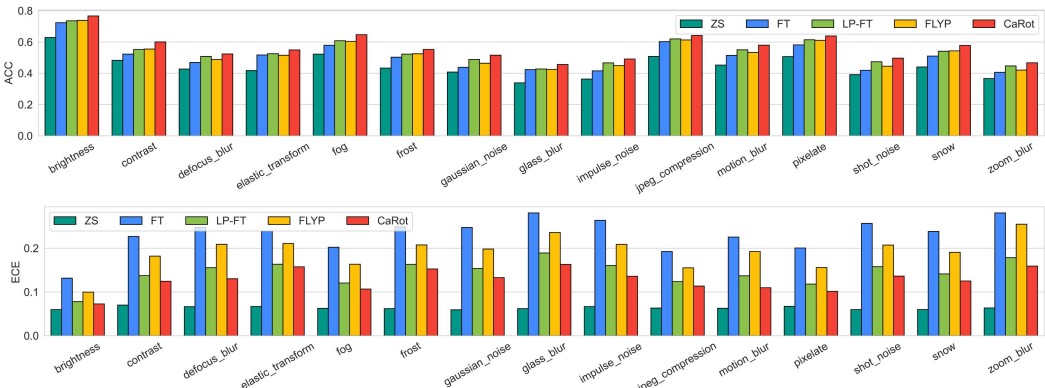

Figure 4: **IN-C corruption-wise accuracy (top) and ECE (bottom).** We evaluate accuracy and ECE over 15 types of image corruption with five corruption severity and report the average performance per corruption. CaRot consistently outperforms baseline methods across diverse corruptions.

**Results on synthetic shifts.** In real-world applications, distribution shifts are commonly occurred by sensory noises. To evaluate different fine-tuning methods under such synthetic shifts, we adopt a corrupted version of ImageNet (IN-C) with 15 types of image corruptions over five severities. In Figure 4, we provide corruption-wise accuracy and ECE of each method averaged by five severities. Overall, CaRot consistently outperforms the base-line methods and the actual amount of improve-ments varying depends on the type of corruptions. Specifically, on the relatively coarser granular cor-ruptions such as Snow, Frost, Fog, Brightness, and Contrast greatly change the semantics of the image

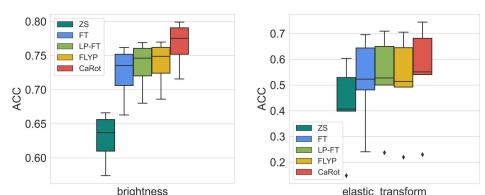

Figure 5: **Closer look at the effectiveness of CaRot on different corruptions.** We provide IN-C accuracy on brightness (left) and elastic transform (right) cor-ruptions. CaRot excels on the coarser corruption such as brightness whereas its effectiveness is weakened on the finer corruption such as elastic transform.

(similar to natural shift), CaRot shows remarkably good performance compared to others. Meanwhile, on the finer granular corruptions such as Elastic transform and JPEG compression, the improvements achieved by CaRot become smaller. We present zoom-in results on these two cases in Figure 5.

### 5.3 Further empirical studies

**Ablation study.** In Table 4, we provide results of the ablation study to show the impacts of each component of CaRot. In line with our hypothesis, results confirm that all three components boost OOD accuracy and calibration performance. The comparison of adopting and not adopting $\mathcal{L}_{\mathrm{MCL}}$ (we followed the naive fine-tuning approach for the latter) ascertains that employing contrastive loss as a fine-tuning objective is superior to cross-entropy loss for ID/OOD accuracy, consistent with the previous observations [17], and even extends to improvements in calibration as well. The ablations of

---

[3]We report the average accuracy over four shifted datasets in Table 6. Results with an asterisk (*) are taken from the original papers. ECE values are not included due to the missing evaluations from the original papers.

Table 4: **Ablation study on CaRot components.** We report accuracy and ECE on ImageNet (ID) and its distribution shifts (OOD). OOD values are averaged over five shifts. Values in brackets indicate the performance difference compared to the first row of each sub-table, and the dark green highlights the positive improvement.

| $\mathcal{L}_{MCL}$ | $\mathcal{L}_{OC}$ | $\mathcal{L}_{SD}$ | ID Acc.↑ | ID ECE↓ | OOD Acc.↑ | OOD ECE↓ |
|---|---|---|---|---|---|---|
| - | - | - | 81.53 | 0.0884 | 57.50 | 0.2186 |
| - | ✓ | - | 81.45 (-0.08) | 0.0874 (-0.0010) | 59.10 (+1.60) | 0.2051 (-0.0135) |
| - | - | ✓ | 82.18 (+0.65) | 0.0601 (-0.0283) | 60.73 (+3.23) | 0.1698 (-0.0488) |
| - | ✓ | ✓ | 82.20 (+0.67) | 0.0634 (-0.0250) | 60.11 (+2.61) | 0.1762 (-0.0424) |
| ✓ | - | - | 82.69 | 0.0635 | 59.40 | 0.1836 |
| ✓ | ✓ | - | 82.51 (-0.18) | 0.0651 (+0.0016) | 59.51 (+0.11) | 0.1803 (-0.0033) |
| ✓ | - | ✓ | 83.03 (+0.34) | 0.0523 (-0.0112) | 62.28 (+2.88) | 0.0772 (-0.1064) |
| ✓ | ✓ | ✓ | 83.13 (+0.44) | 0.0470 (-0.0165) | 62.55 (+3.15) | 0.0791 (-0.1045) |

Table 5: **Analysis on coefficient terms of CaRot objective**. Along with Table 4, we report fine-grained analysis results on each term. We set $\lambda_{OC}$ as 0.2 and $\lambda_{SD}$ as 1.5 when ablating each other and for all experiments throughout the paper. We select the final values of $\lambda_{OC}$ and $\lambda_{SD}$ based on ID ECE and $\sigma_{min}(\tilde{\Sigma}_{\mathcal{D}_{ID}})$, respectively.

| | ID | | OOD | | | ID | | OOD | |
|---|---|---|---|---|---|---|---|---|---|
| $\lambda_{OC}$ | Acc.↑ | ECE↓ | Acc.↑ | ECE↓ | $\lambda_{SD}$ | Acc.↑ | ECE↓ | Acc.↑ | ECE↓ |
| 0.0 | 83.03 | 0.0523 | 62.28 | 0.0772 | 0.0 | 82.51 | 0.0651 | 59.51 | 0.1803 |
| 0.1 | 83.18 | 0.0511 | 62.42 | 0.0779 | 0.5 | 83.07 | 0.0482 | 61.38 | 0.1377 |
| 0.2 | 83.13 | 0.0470 | 62.55 | 0.0791 | 1.0 | 83.23 | 0.0388 | 62.21 | 0.0997 |
| 0.5 | 83.04 | 0.0478 | 62.44 | 0.0798 | 1.5 | 83.13 | 0.0470 | 62.55 | 0.0791 |
| 1.0 | 83.09 | 0.0499 | 62.49 | 0.0781 | 2.0 | 82.72 | 0.0634 | 62.54 | 0.0781 |

the orthogonality constraint and adopting self-distillation validate our rationale behind adding the terms to our learning objective, where we expect them to lower the upper bound of OOD classification and calibration errors. Together, constraining the singular values of image representation on MCL and distilling EMA teacher's predictions show the best results which aligned with results from the demonstration of error bounds in Fig 3. We speculate that learning diverse features by the singular value regularization while being enforced to contribute to reflecting the in-batch similarity structure by EMA SD induces well-restricted solution space [25] otherwise has risk converged to bad solutions (learning diverse features but noise-sensitive). Besides, Table 5 shows the impact of each component by varying the strength coefficients. We observe that the increased intensity of constraint improves ID and OOD performance to some degree, but there is a slight decline when the intensity is too strong. Meanwhile, the strength of self-distillation positively correlated with OOD accuracy and ECE, but there are negative effects on ID accuracy and ECE, which reflects the inevitable trade-off between ID adaptation and OOD generalization (Table G and F in Appendix provide further details).

Table 6: **ImageNet Acc. (except ObjectNet) with additional baselines.**

| Method | IN↑ | IN-V2↑ | IN-R↑ | IN-A↑ | IN-S↑ | Avg. shifts↑ |
|---|---|---|---|---|---|---|
| ZS | 68.33 | 61.93 | 77.71 | 49.95 | 48.26 | 59.46 |
| FT | 81.53 | 71.66 | 70.14 | 44.01 | 49.11 | 58.73 |
| LP-FT | 82.17 | 72.06 | 70.47 | 46.29 | 48.68 | 59.38 |
| CAR-FT* | 83.30 | 74.00 | 75.40 | 49.50 | 53.00 | 62.98 |
| FLYP | 82.69 | 72.73 | 71.35 | 48.52 | 49.84 | 60.61 |
| Lipsum-FT | 83.30 | 73.60 | 75.90 | 49.90 | 51.40 | 62.70 |
| Model Stock* | 84.10 | 74.80 | 71.80 | 51.20 | 51.80 | 62.40 |
| ARF* | 82.70 | 72.80 | 75.60 | 50.30 | 51.80 | 62.63 |
| CaRot (Ours) | 83.13 | 74.11 | 77.71 | 51.60 | 52.71 | 64.03 |

Figure 6: **Impact of $\mathcal{L}_{MCL\text{-}con}$**

**Analysis on singular values.** Figure 6 illustrates the last 20 singular values of the covariance matrix $\bar{I}^T \bar{I}$ where $\bar{I}$ is a standardized image representations over $N$ samples. Our proposed constrained contrastive loss $\mathcal{L}_{MCL\text{-}con}$ increases the small singular values compared to the vanilla contrastive loss $\mathcal{L}_{MCL}$. This result verifies that adding the orthogonality constraint successfully reduces $1/\sigma_{min}(\tilde{\Sigma}_{\mathcal{D}_{ID}})$, the component of the shared upper bound we derived in §3, following our intention.

Table 7: **ImageNet accuracy and ECE on different backbones**. We provide summarized results on CLIP RN50 and ViT-L/14. The best and the second-best in each column are underlined. (See Table H and I for details.)

| | Method | ID Acc.↑ | ID ECE↓ | OOD Acc.↑ | OOD ECE↓ | | ID Acc.↑ | ID ECE↓ | OOD Acc.↑ | OOD ECE↓ |
|---|---|---|---|---|---|---|---|---|---|---|
| RN50 | ZS | 59.83 | 0.0624 | 42.52 | 0.0955 | ViT-L/14 | 75.55 | 0.0590 | 70.93 | 0.0711 |
| | FT | 76.21 | 0.0983 | 41.97 | 0.2804 | | 85.26 | 0.0993 | 65.98 | 0.2036 |
| | LP-FT | 76.25 | 0.1042 | 41.62 | 0.3274 | | 84.74 | 0.1056 | 64.11 | 0.2521 |
| | FLYP | 76.16 | 0.0516 | 42.70 | 0.2127 | | 86.19 | 0.0729 | 71.44 | 0.1470 |
| | CaRot (Ours) | 76.12 | 0.0471 | 42.71 | 0.1714 | | 86.95 | 0.0349 | 74.13 | 0.0737 |

# 6 Related Work

**Robust fine-tuning for visual foundation models.** Beyond the ID generalization, there are a lot of works aiming at improving the generalization of fine-tuned models on the OOD domain. Some of them leverage the strong robustness of pre-trained model through weight-average [61, 26, 53, 44] or regularization [53, 57, 58, 42] whereas others attribute to the robustness during fine-tuning from different part of model backbone [30, 32]. Besides, Goyal et al. [17] claims that aligning the learning objective during pre-training and fine-tuning is crucial for retaining the remarkable OOD generalization capability of the pre-trained model. Although the above methods have provided insights into the extrapolation of foundation models regarding accuracy, confidence calibration has been unexplored, which is crucial for reliable ML applications. We investigate the OOD calibration of fine-tuned CLIP as well as accuracy and propose a unified fine-tuning strategy with theoretical support to achieve superior ID and OOD calibration alongside OOD generalization for the first time.

**Confidence calibration.** After some early research on calibrated prediction [39, 8], lots of follow-up studies have been conducted. As a seminal work, Guo et al. [18] revealed the miscalibration problem of neural networks, then, Minderer et al. [35] and LeVine et al. [33] provided a comprehensive analysis on the calibration of modern vision models with consideration on distribution shift. To improve the calibration of predictive models, Temperature Scaling (TS) [18] and Label Smoothing (LS) [55] are two representative methods in practice. TS-based approaches learn a temperature parameter itself [18, 16] or model [63, 28] to estimate the temperature to adjust the output probability of models, whereas LS-based methods focus on producing soft labels to mitigating overconfidence issues by a fixed [55, 38], randomized [56], or model-based [71, 66] smoothing strategies. However, existing approaches do not consider distribution shifts [71], assume accessibility to target domain [16, 63], assume specific type of distribution shift [59], cannot adjust confidences individually [18, 55]. In this work, we adopt EMA self-distillation as an effective input-dependent calibration method and show that the superior calibration results on in-domain samples can be transferred to other domains (without data from those domains) by pursuing the larger smallest singular value together.

# 7 Conclusion and Discussion

While there have been numerous research endeavors to improve reliability during the model adaptation in the wilds, almost all of them meet the desired criteria only in half: OOD generalization or confidence calibration. This work attempts to address both OOD generalization and OOD calibration in a single framework. We first derive a shared upper bound for OOD classification and calibration errors which is constructed with the ID calibration error and the smallest singular value of ID input representation. We then devise a novel fine-tuning method CaRot, which promotes a larger smallest singular value and calibrated prediction through constrained multimodal contrastive loss and self-distillation. Our theoretical statements and proposed method are empirically validated through extensive experiments.

**Limitation and future work.** Due to resource constraints, our research reached the scale of ViT-L. Exploring validation on the larger models such as ViT-G or ViT-H where the assumptions behind our theory become more realistic would be necessary. Our scope of validation was also limited to CLIP-like VLMs, but Theorem 3 is not specific to VLMs, and investigating the applicability to other types of models such as language models would be an exciting future work direction.

**Impact statement.** Our method enhances the foundation models' reliability in multiple dimensions – accuracy and confidence calibration, and many downstream applications for society can enjoy benefits from the improved reliability. However, inherent biases learned from ID data can not be removed with our method, and thus may have risk raising potential harms in real-world applications.

## Acknowledgement

We thank the NeurIPS24 reviewers, Soojeong Lee and Jinho Kang, for their constructive feedback. Some parts of experiments are based on the NAVER Smart Machine Learning (NSML) platform [54]. This work was supported by Institute of Information & communications Technology Planning & Evaluation (IITP) grant funded by the Korea government (MSIT) (RS-2022-00143911, AI Excellence Global Innovative Leader Education Program) and (No.RS-2019-II190075 Artificial Intelligence Graduate School Program(KAIST)), and the National Research Foundation of Korea(NRF) grant funded by the Korea government(MSIT)(RS-2024-00457216, 2022R1A4A3033874). This work was also supported by the Air Force Research Laboratory under agreement number FA8750-19-2-0200, and by grants from the Defense Advanced Research Projects Agency (DARPA) through the GAILA program (award HR00111990063) and the AIDA program (FA8750-18-20018). The U.S. Government is authorized to reproduce and distribute reprints for governmental purposes notwithstanding any copyright notice herein. The views and conclusions expressed in this document are those of the authors and should not be interpreted as representing the official policies or endorsements, either expressed or implied, of the Air Force Research Laboratory or the U.S. Government.

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

# Appendix

## A  Experimental Detail

This section supplements §5 by providing detailed descriptions for experiments to enhance reproducibility.

### A.1  Details for numerical analysis on error bounds

In §5.1, we conducted toy experiments to perform empirical analyses that demonstrate our theoretical findings. We provide the details of the toy experiments using synthetic data.

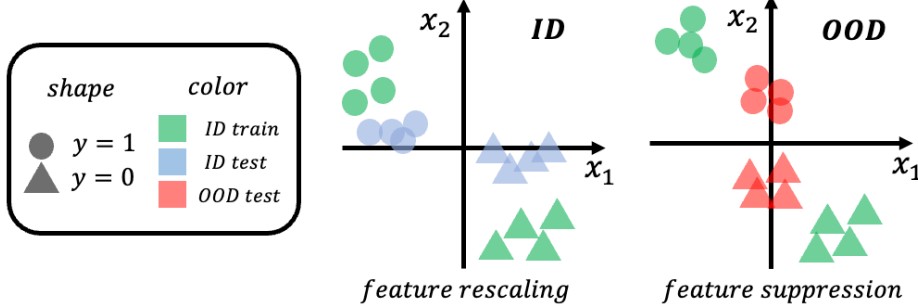

Figure A: Two-dimensional illustration of the experimental setup for numerical analyses. Note that the actual number of dimensions used for the experiments is set to 1000.

We generate 1000-dimensional Gaussian random variables, where the variables have binary noisy labels (15% of random flip) for the ID train, ID test, and OOD test datasets. For the ID train set, the first 400 dimensions and the second 400 dimensions are correlated with labels, and the remaining 200 dimensions are zero-centered random noises. We build the OOD test set from the ID test set by shifting the mean of the first 400 dimensions and downscaling the second 400 dimensions in half. The remaining 200 dimensions are intact. For example, the feature $x_2$ in Figure A is perfectly correlated with labels across train and test in both ID and OOD environments, while the correlation between feature $x_1$ and labels vanish in OOD environment. We train the three-layer multi-layer perceptron networks with four different learning objectives adopting regularization terms for calibration ($\mathcal{L}_{\text{SD}}$) and for the smallest singular value ($\mathcal{L}_{\text{OC}}$) with varying regularization magnitudes (111 models in total): (i) without regularization (Baseline) (ii) with $\mathcal{L}_{\text{SD}}$, (iii) with $\mathcal{L}_{\text{OC}}$, and (iv) with $\mathcal{L}_{\text{SD}}$ and $\mathcal{L}_{\text{OC}}$. For $\mathcal{L}_{\text{OC}}$, we use an orthogonal constraint over the last weight matrix. For $\mathcal{L}_{\text{SD}}$, we adopt Born-Again-Network (BAN)-style self-distillation [13]. After training, we measured $\sigma_{min}$ and ECE on the ID test set and measured the mean squared error (MSE) and ECE on OOD test set. The visualization results of Figure 3 show the correlation between the OOD MSE (or OOD ECE) and its upper bound (average of the standardized[4] smallest singular value of ID feature representation and standardized ID ECE).

### A.2  Benchmark datasets

This section supplements the summarized explanations of datasets provided in §5.2.

Training and test splits of ImageNet-1K [10] consist of 1000 classes, and its variants have the entire 1000 or a subset of the classes. Following Radford et al. [47] and Goyal et al. [17], we use the OpenAI templates to create text descriptions for each class (80 templates per class) for evaluation, and the averaged text representation is used as the final class representation for evaluation. Several related datasets including ImageNet-V2 [49], ImageNet-Rendition [23], ImageNet-A [24], ImageNet-Sketch [60], and ObjectNet [3] are employed to evaluate robustness of models. These datasets consist of similar semantic classes but are collected from diffrent input distributions or styles.

---

[4]computed by subtracting the mean and dividing with standard deviation of all 111 models' metric values

## A.3 Baseline methods

This section supplements the summarized explanations of baseline methods discussed in §5.2.

**Zero-shot (ZS [47])**: Zero-shot classifier is obtained by encoding and averaging text representations of each class using the pre-trained CLIP text encoder.

**Standard fine-tuning (FT [61])**: Linear classification head is initialized with text representation vectors for each class encoded by pre-trained CLIP text encoder. Image encoder and linear classifier parameters are fine-tuned for 10 epochs with a learning rate of 3e-5.

**LP-FT [30]**: Randomly initialized linear classification head is first trained upon frozen image encoder for 5 epochs and then both image encoder and linear head parameters are updated for 5 epochs. For each phase, we use a learning rate of 1e-2 and 3e-5, respectively.

**Fine-tuning with contrastive learning (FLYP [17])**: Both image and text encoders are updated without additional linear classification heads. To create text representations of training samples, we use the OpenAI template similar to the evaluation data. Unlike for evaluation, we do not take an average of 80 different templates. Instead, we use 80 versions of text prompts for each class to build training pairs. The training pairs are randomly selected throughout the training steps. We fine-tune the model for 10 epochs (in total, 25K steps) with a learning rate of 1e-5.

**Lipsum-FT [42]**: Cross-entropy loss with a regularization term that minimizes the energy gap between image and text is used as fine-tuning objective. We followed the details described in the original paper.

**CaRot (Ours)**: We set the orthogonality constraint coefficient $\lambda_{OC}$ as 0.2 and self-distillation coefficient $\lambda_{SD}$ as 1.5, update frequency for EMA teacher as 500, and EMA final target momentum as 0.9. We linearly increased the EMA momentum $\alpha$ by 0.05 for the first 20% iterations. We followed all the other details from FLYP.

# B    Additional Evaluation Results

In addition to the comparisons of our CaRot with zero-shot, naive fine-tuning, and robust fine-tuning methods in §5, we provide results when applying post-hoc techniques for robustness (weight ensembling) and calibration (temperature scaling). Moreover, we compare our self-distillation-based soft label with uniform constant label smoothing.

## B.1    Comparing approaches

**Weight average of fine-tuned and zero-shot (WiSE-FT [61])**: Zero-shot and fine-tuned model weights are averaged with a strength of ensembling coefficient. This ensembling technique can be applied to any fine-tuning method. We tune the ensembling coefficient for each method based on the ImageNet ECE value (we picked the value having the lowest ID ECE for each method).

**Temperature scaling (TS [18])**: Before applying the softmax function to compute output probability distribution, TS divides the logit by temperature $\tau \in (0, \infty)$. $\tau \to 0$ makes the probability similar to point masses (sharpening), $\tau \to \infty$ makes uniform distribution (smoothing). Scaling the output distribution does not affect accuracy since it does not change the model prediction (i.e., the probability rank remains the same). Temperature value was tuned for each method on the ID validation set based on the ECE value.

**Label smoothing (LS [55])**: LS is a regularization strategy that pursues the generalization of classification by utilizing soft labels, which are derived by adding uniform distribution to the hard label distribution. The soft label can be viewed as a new target probability distribution where the value of 1 to the target pair is reduced and the value of 0 for the non-target pair is increased by the smoothing parameter $\epsilon \in (0, 1)$. Since utilizing soft labels allows the model to pull negative pairs with limited strength, LS is beneficial for calibration beyond generalization by addressing the over-confidence issue [38].

## B.2  Additional result and discussion

**Results with WiSE** are reported in Table B and C. We observe that weight ensembling, which aims to align zero-shot models and fine-tuned models in the model weight space, boosts the overall performance. Still, CaRot shows superior results to the baselines.

**Results with TS** can be found in Table D. Aligning with previous observations, applying TS significantly improves ID ECE. However, since TS adjusts the logits assuming that all data instances would have a similar extent of overconfidence (or underconfidence) issue, its positive effect is limited under distribution shifts. Hence, TS does not guarantee to attain low calibration error on OOD datasets. It is noteworthy that applying TS even harms OOD ECE of CaRot (compare results with Table 3), which is already calibrated using data-dependent confidence adjustment during fine-tuning.

**Results comparing with LS** are reported in Table E. In accordance with its capability of improving calibration, LS successfully reduces calibration errors. We observe that our adopted approach, EMA SD, remarkably outperforms LS, especially on OOD ECE as well as in ID ECE. While LS addresses the over-confidence problem during train time by dispersing the concentrated confidence on the target label to non-target labels with a constant amount, self-distillation shows a similar behavior but in a dynamic approach, considering the diversity of data instances [71]. We interpret that reflecting the difficulty of input batch and distilling such information is crucial to achieving robust calibration.

We further provide an intuitive interpretation of self-distillation as an input-dependent approach to label smoothing as elaborated in Zhang and Sabuncu [71]. EMA SD provides soft labels considering the variation of data instances. For example, classifying a `dog` image from images of `airplane` or `car` could be less challenging than classifying it from images of `cat` or `wolf`. Ideally, a calibrated model should output higher confidence for the former case than for the latter. Instead of providing supervision of constantly adjusted confidence (as in LS), teacher predictions provide confidence reflecting the input data difficulty. Thereby, the student model can be supervised with high confidence diversity, which leads to a better-calibrated model. Please refer to Zhang and Sabuncu [71] for detailed discussions.

**Other types of regularization for larger smallest singular value.** We adopted an orthogonality constraint over visual projection matrix $W_v \in \mathbb{R}^{d_v \times r}$ via $||W_v^T W_v - \mathbf{I}||_F$ term which pursues the larger effective rank of the visual projection matrix. While we adopt this kind of indirect soft constraint to achieve balanced performance over both ID and OOD generalization, one may wonder about 1) the implication of increased effective rank of $W_v$ on the smallest singular value of the covariance matrix and 2) the possibility of leveraging more direct constraints to increase the smallest singular value of the input covariance matrix. This paragraph answers those questions.

Given $n$ samples of $d$-dimensional visual features and $r$ as a pre-defined projection dimension, let $\tilde{Z}_v \in \mathbb{R}^{n \times r}$ and $\hat{Z}_v \in \mathbb{R}^{n \times r}$ denote visual representations obtained by an arbitrary projection matrix $W_v \in \mathbb{R}^{d_v \times r}$ and an orthogonal projection matrix $O_v \in \mathbb{R}^{d_v \times r}$ multiplied with pre-projected visual feature $Z_v \in \mathbb{R}^{n \times d_v}$, respectively. We can show that the rank of the covariance matrix from $\hat{Z}_v$ is always greater or equal to that of $\tilde{Z}_v$ as below,

$$\text{rank}(\tilde{Z}_v^T \cdot \tilde{Z}_v) = \text{rank}(\tilde{Z}_v) \tag{6}$$
$$= \text{rank}(Z_v \cdot W_v) \tag{7}$$
$$\leq \text{rank}(Z_v \cdot O_v) \tag{8}$$
$$= \text{rank}(\hat{Z}_v) \tag{9}$$
$$= \text{rank}(\hat{Z}_v^T \cdot \hat{Z}_v) \tag{10}$$

During minimizing $||W_v^T W_v - \mathbf{I}||_F$, the visual projection matrix $W_v$ becomes closer to an orthogonal matrix $O_v$, and the effective rank of the visual representations' covariance matrix increases. A larger effective rank implies a non-diminishing (relatively larger) smallest singular value of a matrix, which justifies our implementation of the method.

Meanwhile, we can adopt other constraint terms alternative to orthogonality constraint over visual projection matrix. Using the negative value of the smallest singular value of the visual representation's covariance matrix as an additional loss term may be the most natural candidate. The result is provided in Table B.2. While both SVD and orthogonality constraint methods show remarkably better performance in terms of OOD classification and calibration, SVD requires much heavier

computation compared with the orthogonality constraint term. For simplicity, we advocate the use of orthogonality term thus. Exploring other types of efficient constraint terms will be a promising future work direction.

Table A: Comparison between SVD-based regularization and the orthogonality constraint term. Both terms are effective in terms of OOD generalization and calibration, but SVD requires a much heavier computation.

| | | ID | | OOD | | ID-OOD Gap | |
| --- | --- | --- | --- | --- | --- | --- | --- |
| | Time Complexity | Acc. ($\uparrow$) | ECE ($\downarrow$) | Acc. ($\uparrow$) | ECE ($\downarrow$) | Acc. ($\downarrow$) | ECE ($\downarrow$) |
| FLYP | - | 82.69 | 0.0635 | 59.40 | 0.1836 | 23.29 | 0.1201 |
| CaRot (SVD) | $\mathcal{O}(D^2N + D^3)$ | 83.05 | 0.0536 | 62.40 | **0.0770** | 20.65 | **0.0234** |
| CaRot (Ours) | $\mathcal{O}(D^2N + D^2)$ | **83.13** | **0.0470** | **62.55** | 0.0791 | **20.58** | 0.0321 |

Table B: **Accuracy on ImageNet and distribution shifts using WiSE-FT [61]**. We select the optimal ensembling coefficient (i.e., $\alpha$) for each method.

| Method | IN$\uparrow$ | IN-V2$\uparrow$ | IN-R$\uparrow$ | IN-A$\uparrow$ | IN-S$\uparrow$ | ObjectNet$\uparrow$ | Avg. shifts$\uparrow$ |
| --- | --- | --- | --- | --- | --- | --- | --- |
| ZS | 68.33 | 61.93 | 77.71 | 49.95 | 48.26 | 54.17 | 58.39 |
| FT | 81.96 | 72.69 | 77.19 | 51.93 | 53.17 | 56.83 | 62.36 |
| LP-FT | 82.63 | 73.14 | 75.24 | 51.92 | 51.99 | 55.86 | 61.63 |
| FLYP | 82.53 | 73.65 | 77.57 | 54.65 | 53.23 | 58.02 | 63.42 |
| CaRot | 82.36 | 73.72 | 79.58 | 54.07 | 53.96 | 57.70 | 63.81 |

Table C: **ImageNet ECE results with WiSE-FT [61]**. Along with Table B, we report the ECE results.

| Method | IN$\downarrow$ | IN-V2$\downarrow$ | IN-R$\downarrow$ | IN-A$\downarrow$ | IN-S$\downarrow$ | ObjectNet$\downarrow$ | Avg. shifts$\downarrow$ |
| --- | --- | --- | --- | --- | --- | --- | --- |
| ZS | 0.0570 | 0.0548 | 0.0541 | 0.0967 | 0.0850 | 0.0780 | 0.0736 |
| FT | 0.0714 | 0.0873 | 0.0744 | 0.1509 | 0.1391 | 0.1528 | 0.1209 |
| LP-FT | 0.0510 | 0.0895 | 0.0561 | 0.1917 | 0.1587 | 0.2014 | 0.1395 |
| FLYP | 0.0773 | 0.1087 | 0.0806 | 0.1963 | 0.1798 | 0.1995 | 0.1530 |
| CaRot | 0.0427 | 0.0416 | 0.0490 | 0.1207 | 0.0731 | 0.1113 | 0.0791 |

## B.3 Detailed results from main paper

**Coefficient terms ablation.** In Table G and F, we present the detailed ablation results of hyperparameters associated with the methodologies addressed in our paper. These results supplement Table 5.

**Different VLM backbones.** We provide the full results of Table 7 in Table H and I.

**Visualization of ECE values.** Figure B illustrates the reliability diagram of ImageNet ECE results reported in Table 3.

Table D: **ImageNet ECE results with temperature scaling (TS)**. Supplement to Table 2 and 3, we provide results applying TS. Note that TS is a post-hoc method and does not affect accuracy. The temperature is selected using IN ECE for each method.

| Method | IN↓ | IN-V2↓ | IN-R↓ | IN-A↓ | IN-S↓ | ObjectNet↓ | Avg. shifts↓ |
|---|---|---|---|---|---|---|---|
| ZS | 0.0392 | 0.0633 | 0.0532 | 0.1792 | 0.1370 | 0.1760 | 0.1217 |
| FT | 0.0463 | 0.0786 | 0.0484 | 0.1798 | 0.1408 | 0.1820 | 0.1259 |
| LP-FT | 0.0382 | 0.0509 | 0.0477 | 0.1450 | 0.1028 | 0.1433 | 0.0979 |
| FLYP | 0.0392 | 0.0633 | 0.0532 | 0.1792 | 0.1370 | 0.1760 | 0.1217 |
| Lipsum-FT | 0.0380 | 0.0599 | 0.0419 | 0.1445 | 0.1165 | 0.1362 | 0.0998 |
| CaRot | 0.0401 | 0.0527 | 0.0437 | 0.1520 | 0.0802 | 0.1373 | 0.0931 |

Table E: **Comparison on LS and EMA SD**. We compare the impact of LS and EMA SD with $\mathcal{L}_{\text{MCL-con}}$ as calibration regularization.

| Method | | | | Acc.↑ | | | |
|---|---|---|---|---|---|---|---|
| | IN | IN-V2 | IN-R | IN-A | IN-S | ObjectNet | Avg. shifts |
| – | 82.51 | 73.18 | 71.80 | 48.16 | 49.78 | 54.67 | 59.51 |
| LS | 82.53 | 73.33 | 71.90 | 48.33 | 49.46 | 54.99 | 59.60 |
| EMA SD | 83.13 | 74.11 | 77.71 | 51.60 | 52.71 | 56.60 | 62.55 |
| | | | | ECE↓ | | | |
| – | 0.0651 | 0.1104 | 0.0910 | 0.2459 | 0.2132 | 0.2411 | 0.1803 |
| LS | 0.0475 | 0.0726 | 0.0526 | 0.1534 | 0.1533 | 0.1993 | 0.1262 |
| EMA SD | 0.0470 | 0.0367 | 0.0575 | 0.1240 | 0.0699 | 0.1075 | 0.0791 |

Table F: **Orthogonality constraint hyperparameter.** We report the impact of the orthogonality constraint term of the CaRot objective by ablating its strength coefficient $\lambda_{\text{OC}}$. We set our final value as 0.2, tuning based on ID ECE.

| $\lambda_{\text{OC}}$ | | | | Acc.↑ | | | |
|---|---|---|---|---|---|---|---|
| | IN | IN-V2 | IN-R | IN-A | IN-S | ObjectNet | Avg. shifts |
| 0.1 | 83.18 | 74.10 | 77.53 | 51.35 | 52.66 | 56.47 | 62.42 |
| 0.2 | 83.13 | 74.11 | 77.71 | 51.60 | 52.71 | 56.60 | 62.55 |
| 0.5 | 83.04 | 74.40 | 77.64 | 51.04 | 52.63 | 56.49 | 62.44 |
| 1.0 | 83.09 | 74.35 | 77.59 | 51.23 | 52.65 | 56.62 | 62.49 |
| $\lambda_{\text{OC}}$ | | | | ECE↓ | | | |
| 0.1 | 0.0511 | 0.0382 | 0.0620 | 0.1190 | 0.0712 | 0.0990 | 0.0779 |
| 0.2 | 0.0470 | 0.0367 | 0.0575 | 0.1240 | 0.0699 | 0.1075 | 0.0791 |
| 0.5 | 0.0478 | 0.0408 | 0.0579 | 0.1253 | 0.0701 | 0.1048 | 0.0798 |
| 1.0 | 0.0499 | 0.0380 | 0.0609 | 0.1201 | 0.0693 | 0.1022 | 0.0781 |

Table G: **Self-distillation term hyperparameter.** We report the impact of the EMA SD term of CaRot objective by ablating its strength coefficient $\lambda_{\text{SD}}$. We set our final value as 1.5, tuning based on $\sigma_{min}$.

| $\lambda_{\text{SD}}$ | | | | Acc.↑ | | | |
|---|---|---|---|---|---|---|---|
| | IN | IN-V2 | IN-R | IN-A | IN-S | ObjectNet | Avg. shifts |
| 0.5 | 83.07 | 74.22 | 74.37 | 50.76 | 51.49 | 56.08 | 61.38 |
| 1.0 | 83.23 | 74.51 | 76.38 | 51.05 | 52.47 | 56.63 | 62.21 |
| 1.5 | 83.03 | 74.13 | 77.59 | 50.72 | 52.49 | 56.49 | 62.28 |
| 2.0 | 82.72 | 74.14 | 78.25 | 50.71 | 53.13 | 56.49 | 62.54 |
| $\lambda_{\text{SD}}$ | | | | ECE↓ | | | |
| 0.5 | 0.0482 | 0.0791 | 0.0599 | 0.2002 | 0.1533 | 0.1960 | 0.1377 |
| 1.0 | 0.0388 | 0.0544 | 0.0405 | 0.1640 | 0.0914 | 0.1481 | 0.0997 |
| 1.5 | 0.0523 | 0.0401 | 0.0642 | 0.1173 | 0.0732 | 0.0910 | 0.0772 |
| 2.0 | 0.0634 | 0.0467 | 0.0785 | 0.1030 | 0.0796 | 0.0829 | 0.0781 |

Table H: **ImageNet results on CLIP ResNet50**

| Method | IN | IN-V2 | IN-R | IN-A | IN-S | ObjectNet | Avg. shifts |
|--------|-----|-------|------|------|------|-----------|-------------|
| | | | | Acc.↑ | | | |
| ZS | 59.83 | 52.90 | 60.72 | 23.25 | 35.45 | 40.27 | 42.52 |
| FT | 76.21 | 64.87 | 50.66 | 18.11 | 33.90 | 42.32 | 41.97 |
| LP-FT | 76.25 | 64.48 | 49.55 | 18.60 | 33.33 | 42.13 | 41.62 |
| FLYP | 76.16 | 65.10 | 51.55 | 20.08 | 34.24 | 42.53 | 42.70 |
| CaRot (Ours) | 76.12 | 65.36 | 52.16 | 19.32 | 34.05 | 42.67 | 42.71 |
| | | | | ECE↓ | | | |
| ZS | 0.0624 | 0.0559 | 0.0530 | 0.2048 | 0.0740 | 0.0899 | 0.0955 |
| FT | 0.0983 | 0.1623 | 0.1860 | 0.4692 | 0.2824 | 0.3023 | 0.2804 |
| LP-FT | 0.1042 | 0.1759 | 0.2709 | 0.5184 | 0.3520 | 0.3197 | 0.3274 |
| FLYP | 0.0516 | 0.0872 | 0.1439 | 0.3872 | 0.2021 | 0.2432 | 0.2127 |
| CaRot (Ours) | 0.0471 | 0.0601 | 0.0948 | 0.3435 | 0.3435 | 0.2127 | 0.1714 |

Table I: **ImageNet results on CLIP ViT-L/14**

| Method | IN | IN-V2 | IN-R | IN-A | IN-S | ObjectNet | Avg. shifts |
|--------|-----|-------|------|------|------|-----------|-------------|
| | | | | Acc.↑ | | | |
| ZS | 75.55 | 69.85 | 87.85 | 70.76 | 59.61 | 66.59 | 70.93 |
| FT | 84.74 | 75.32 | 75.36 | 55.65 | 54.44 | 59.76 | 64.11 |
| LP-FT | 85.26 | 76.76 | 80.21 | 55.95 | 56.84 | 60.12 | 65.98 |
| FLYP | 86.19 | 78.21 | 83.81 | 68.85 | 60.20 | 66.15 | 71.44 |
| CaRot (Ours) | 86.95 | 79.28 | 87.96 | 72.68 | 62.66 | 68.05 | 74.13 |
| | | | | ECE↓ | | | |
| ZS | 0.0590 | 0.0686 | 0.0339 | 0.0640 | 0.1037 | 0.0852 | 0.0711 |
| FT | 0.1056 | 0.1741 | 0.1613 | 0.3151 | 0.3234 | 0.2865 | 0.2521 |
| LP-FT | 0.0993 | 0.1531 | 0.0872 | 0.2593 | 0.2613 | 0.2572 | 0.2036 |
| FLYP | 0.0729 | 0.1219 | 0.0621 | 0.1443 | 0.2164 | 0.1903 | 0.1470 |
| CaRot (Ours) | 0.0349 | 0.0634 | 0.0353 | 0.0732 | 0.0914 | 0.1051 | 0.0737 |

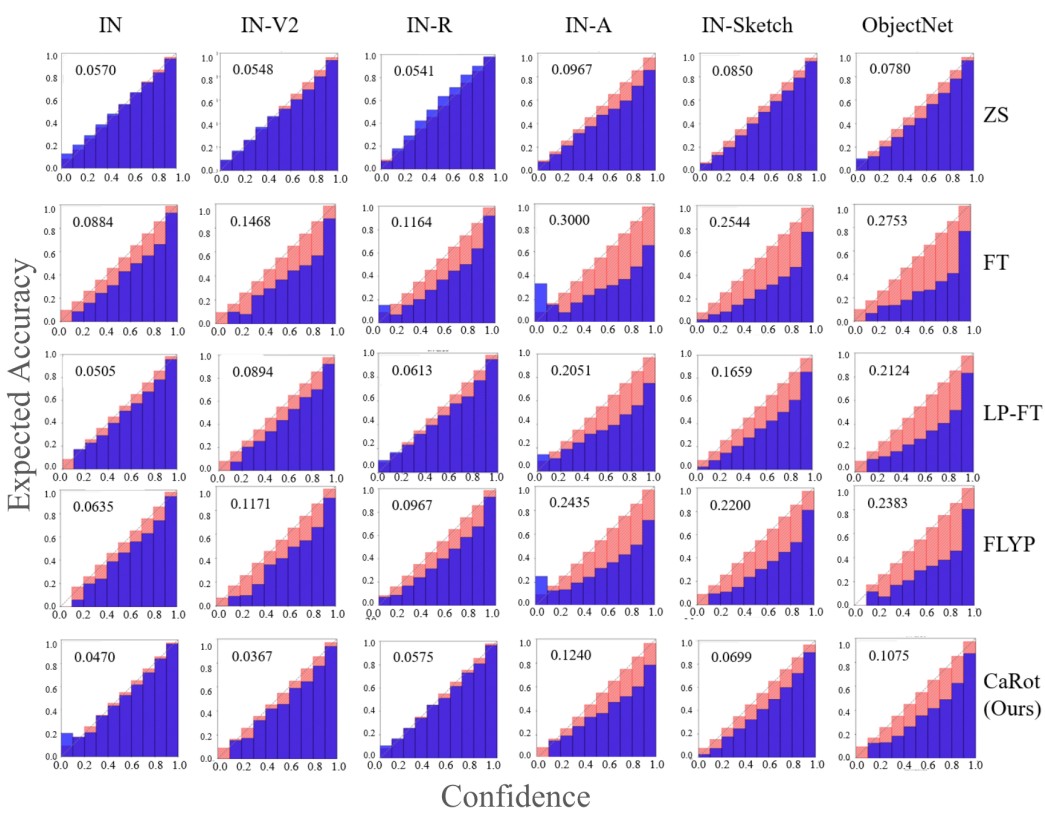

Figure B: **Reliability diagram of ImageNet ECE.** This figure supplements ECE results in Table 3. The value inside each plot indicates ECE.

## C   Proof and further discussion

In this section, we provide proof for Theorem 3 and some discussions.

**Theorem C.1** (Restatement of Theorem 3.). *Let $h : \mathcal{X} \to [0, 1]$ be a real-valued function of structure $h(x) = \sum_{i=1}^{d} h_i(x[i])$ where $h_i$ is an arbitrary one-dimensional function, and $h$ is in a hypothesis class $\mathcal{H}$ that has pseudo dimension $\mathcal{P}dim(\mathcal{H}) = d_h$, $\hat{\mathcal{D}}_{ID}$ be an $N$-size empirical distribution on ID domain. If $(x[1], ..., x[d])$ have matching marginals for ID and OOD, and $(x[i], x[j])$ is a bi-variate Gaussian for every $i, j \in [d]$, then for any $\delta \in (0, 1)$ and for all $h$, the following bounds hold with probability at least $1 - \delta$:*

$$i) \quad \varepsilon_{\mathcal{D}_{OOD}}(h) \leq \varepsilon_{\hat{\mathcal{D}}_{ID}}(h) + \frac{d}{\sigma_{min}(\tilde{\Sigma}_{\mathcal{D}_{ID}})} + \Delta + \mathcal{O}\left(\sqrt{\frac{1}{N} \log\left(\frac{N}{d_h}\right)^{d_h}\left(\frac{1}{\delta}\right)}\right) \tag{11}$$

$$ii) \quad \mathbb{E}_{\mathcal{D}_{OOD}}[(h(x) - y)^2] + \mathbb{E}_{\mathcal{D}_{OOD}}[c(x)^2] - 1 \leq \varepsilon_{\hat{\mathcal{D}}_{ID}}(h) + \frac{d}{\sigma_{min}(\tilde{\Sigma}_{\mathcal{D}_{ID}})} + \Delta + \mathcal{O}\left(\sqrt{\frac{1}{N} \log\left(\frac{N}{d_h}\right)^{d_h}\left(\frac{1}{\delta}\right)}\right) \tag{12}$$

where $\tilde{\Sigma}_{\mathcal{D}_{ID}} := \mathbb{E}_{\mathcal{D}_{ID}}[\tilde{x}\tilde{x}^T]$ is a covariance matrix with a strictly positive minimum singular value of $d$-dimensional normalized input $\tilde{x} = (\tilde{x}[1], ..., \tilde{x}[d])$, where $\tilde{x}[i] := (x[i] - \mathbb{E}[x[i]])\text{Var}(x[i])^{-1/2}$ and $\sigma_{\min}(M)$ is the smallest singular value of a matrix $M \in \mathbb{R}^{d_1 \times d_2}$.

*Proof.* The proof for the above theorem is divided into three steps: 1) the derivation of OOD calibration error bound, 2) the derivation of OOD generalization bound, and 3) the replacement of domain discrepancy term over ID and OOD into an ID-depend singular value term.

We first define a domain discrepancy measure $\mathcal{H}$-sqaure disagreement, $sd_{\mathcal{H}}$, between two distributions as below:

**Definition C.2** ($\mathcal{H}$-sqaure disagreement). *Given two probability distributions $\mathcal{D}_{OOD}$ and $\mathcal{D}_{ID}$ over input space $\mathcal{X}$, $\mathcal{H}$ as a hypothesis class containing the hypothesis $h(\cdot) : \mathcal{X} \to [0, 1]$, the discrepancy between $\mathcal{D}_{OOD}$ and $\mathcal{D}_{ID}$ is defined as*

$$sd_{\mathcal{H}}(\mathcal{D}_{OOD}, \mathcal{D}_{ID}) := \sup_{h,h' \in \mathcal{H}} |\mathbb{E}_{\mathcal{D}_{OOD}}[(h(x) - h'(x))^2] - \mathbb{E}_{\mathcal{D}_{ID}}[(h(x) - h'(x))^2]|. \tag{13}$$

The $\mathcal{H}$-square disagreement can be regarded as a variant of H-divergence [5] which adopts mean square error term rather than $0 - 1$ classification loss or mean absolute error term as in [72].

**Proposition C.3** (OOD calibration error bound). *Let $h : \mathcal{X} \to [0, 1]$ be a real-valued function in a hypothesis class $\mathcal{H}$ with a pseudo dimension $\mathcal{P}dim(\mathcal{H}) = d_h$. If $\hat{\mathcal{D}}_{ID}$ is an empirical distribution constructed by $N$-size i.i.d. samples drawn from $\mathcal{D}_{ID}$, then for any $\delta \in (0, 1)$, and for all $h$, a bound below hold with probability at least $1 - \delta$.*

$$\varepsilon_{\mathcal{D}_{OOD}}(h) \leq \varepsilon_{\hat{\mathcal{D}}_{ID}}(h) + sd_{\mathcal{H}}(\mathcal{D}_{OOD}, \mathcal{D}_{ID}) + \Delta + \mathcal{O}\left(\sqrt{\frac{1}{N}\left(\log\frac{1}{\delta} + d_h \log\frac{N}{d_h}\right)}\right). \tag{14}$$

Now, likewise Lemma 3 of [72], we start to review a triangle inequality for all $h, h', h'' \in \mathcal{H}$, for any $\mathcal{D}$ on $\mathcal{X}$, and for error function $\varepsilon_{\mathcal{D}}(\cdot, \cdot)$, the inequality $\varepsilon_{\mathcal{D}}(h, h') \leq \varepsilon_{\mathcal{D}}(h, h'') + \varepsilon_{\mathcal{D}}(h'', h')$ holds. Here, we use $\varepsilon_D(h, h')$ to denote $\mathbb{E}_{x \sim D}[(h(x) - h'(x))^2]$. Now, given $h^* := \arg\min_{h \in \mathcal{H}} \varepsilon_{\mathcal{D}_{ID}}(h) + \varepsilon_{\mathcal{D}_{OOD}}(h)$ and $\Delta := \varepsilon_{\mathcal{D}_{ID}}(h^*) + \varepsilon_{\mathcal{D}_{OOD}}(h^*)$, we have

$$\begin{aligned}
\varepsilon_{\mathcal{D}_{OOD}}(h) &\leq \varepsilon_{\mathcal{D}_{OOD}}(h^*) + \varepsilon_{\mathcal{D}_{OOD}}(h, h^*) \\
&= \varepsilon_{\mathcal{D}_{OOD}}(h^*) + \varepsilon_{\mathcal{D}_{OOD}}(h, h^*) - \varepsilon_{\mathcal{D}_{ID}}(h, h^*) + \varepsilon_{\mathcal{D}_{ID}}(h, h^*) \\
&\leq \varepsilon_{\mathcal{D}_{OOD}}(h^*) + |\varepsilon_{\mathcal{D}_{OOD}}(h, h^*) - \varepsilon_{\mathcal{D}_{ID}}(h, h^*)| + \varepsilon_{\mathcal{D}_{ID}}(h, h^*) \\
&\leq \varepsilon_{\mathcal{D}_{OOD}}(h^*) + \varepsilon_{\mathcal{D}_{ID}}(h, h^*) + sd_{\mathcal{H}}(\mathcal{D}_{OOD}, \mathcal{D}_{ID}) \\
&\leq \varepsilon_{\mathcal{D}_{OOD}}(h^*) + \varepsilon_{\mathcal{D}_{ID}}(h) + \varepsilon_{\mathcal{D}_{ID}}(h^*) + sd_{\mathcal{H}}(\mathcal{D}_{OOD}, \mathcal{D}_{ID}) \\
&= \varepsilon_{\mathcal{D}_{ID}}(h) + sd_{\mathcal{H}}(\mathcal{D}_{OOD}, \mathcal{D}_{ID}) + \Delta.
\end{aligned} \tag{15}$$

where $\varepsilon_{\mathcal{D}}(h)$ is defined as $\varepsilon_{\mathcal{D}}(h) := \mathbb{E}_{x \sim \mathcal{D}}[(h(x) - h_0(x))^2]$ and $h_0(\cdot) := \arg\min_{h \in \mathcal{H}} \mathbb{E}_x[(h(x) - c(x))^2]$ denotes a desired calibrated predictor of label $y$ when $c(x) := \mathbb{E}_{\mathcal{D}}[y|h(x)]$. Here, the first and fourth inequalities in Eq (15) are held by triangular inequality.

The above bound is defined over the true population distribution $\mathcal{D}_{\text{ID}}$ and $\mathcal{D}_{\text{OOD}}$. For the ID domain, we can confine our analysis to empirical distribution $\hat{\mathcal{D}}_{\text{ID}}$ with $N$ i.i.d. samples generated from $\mathcal{D}_{\text{ID}}$, by leveraging a generalization bound on a single domain regression setting [36, 72]. If $\mathcal{P}dim(\mathcal{H}) = d_h$, for all $h \in \mathcal{H}$, below bound holds with probability at least $1 - \delta$ (See Lemma 5 of Zhao et al. [72]).

$$\varepsilon_{\mathcal{D}_{\text{OOD}}}(h) \leq \varepsilon_{\hat{\mathcal{D}}_{\text{ID}}}(h) + sd_{\mathcal{H}}(\mathcal{D}_{\text{OOD}}, \mathcal{D}_{\text{ID}}) + \Delta + \mathcal{O}\left(\sqrt{\frac{1}{N}\left(\log\frac{1}{\delta} + d_h \log\frac{N}{d_h}\right)}\right). \quad (16)$$

The above bound is similar to a regression bound proposed by Zhao et al. [72], but we build the theory based on the squared error rather than the previously adopted absolute error. This slight change allows us two attractive extensions that we will introduce to achieve calibrated robust fine-tuning.

While Proposition C.3 provides guidance to pursue OOD calibration, it does not say anything about OOD classification error, which is our primary goal before calibration. Here, we pay attention to the decomposition of the Brier score [39, 46]:

$$\underbrace{\mathbb{E}[(h(x) - y)^2]}_{\text{classification error}} = \underbrace{\mathbb{E}[(h(x) - c(x))^2]}_{\text{calibration error}} + 1 - \underbrace{\mathbb{E}[c(x)^2]}_{\text{sharpness}}, \quad (17)$$

where $\mathbb{E}[(h(x) - y)^2]$ is an expected mean-squared classification error and $\mathbb{E}[c(x)^2]$ denotes the *sharpness* [51] term rewarding the predictor $h(\cdot)$ to produce outputs towards zero or one. By assuming that $h(\cdot)$ is expressive enough to estimate the ground truth expectation function for the label, i.e., $c(\cdot)$, plugging Eq. (17) into the LHS of Proposition C.3 derives the same (except a constant) upper bound for the sum of classification error and prediction sharpness on OOD samples as in Proposition C.4.

**Proposition C.4** (OOD generalization error bound). *Let $h(\cdot)$, $\mathcal{H}$, and $\hat{\mathcal{D}}_{\text{ID}}$ have the same definition as in Proposition C.3, then for any $\delta \in (0, 1)$, and for all $h$, a bound hold with prob. at least 1-$\delta$,*

$$\mathbb{E}_{\mathcal{D}_{OOD}}[(h(x) - y)^2] + \mathbb{E}_{\mathcal{D}_{OOD}}[c(x)^2] - 1 \leq \varepsilon_{\hat{\mathcal{D}}_{ID}}(h) + sd_{\mathcal{H}}(\mathcal{D}_{OOD}, \mathcal{D}_{ID}) + \Delta + \mathcal{O}\left(\sqrt{\frac{1}{N}\left(\log\frac{1}{\delta} + d_h \log\frac{N}{d_h}\right)}\right). \quad (18)$$

**From domain discrepancy to minimum singular value.** The second term $sd_{\mathcal{H}}(\mathcal{D}_{\text{OOD}}, \mathcal{D}_{\text{ID}})$ of RHS of Proposition C.3 and Proposition C.4 is defined on both ID and OOD samples, so it is hard to control the quantity directly. While existing approaches attempt to learn domain-invariant representations for reducing the similar quantity (e.g., H-divergence) by using unlabeled OOD data [14, 72, 73], the OOD data are commonly inaccessible on many real-world applications. Therefore, we need a quantity that is solely defined with ID data. Recently, Dong and Ma [11] proved that the domain discrepancy ratio can be bounded from above by a reciprocal of the smallest singular value of a covariance matrix of ID data as below:

$$\sup_{h, h' \in \mathcal{H}} \frac{\mathbb{E}_{\mathcal{D}_{\text{OOD}}}(h(x) - h'(x))^2}{\mathbb{E}_{\mathcal{D}_{\text{ID}}}(h(x) - h'(x))^2} \leq \frac{d}{\sigma_{\min}(\tilde{\Sigma}_{\mathcal{D}_{\text{ID}}})}, \quad (19)$$

where $\tilde{\Sigma}_{\mathcal{D}_{\text{ID}}} := \mathbb{E}_{\mathcal{D}_{\text{ID}}}[\tilde{x}\tilde{x}^T]$ is a covariance matrix of the $d$-dimensional nomarlized input $\tilde{x} = (\tilde{x}[1], ..., \tilde{x}[d])$ where $\tilde{x}[i] := (x[i] - \mathbb{E}[x[i]])\text{Var}(x[i])^{-1/2}$ and $\sigma_{\min}(M)$ is the minimum singular value of a matrix $M \in \mathbb{R}^{d_1 \times d_2}$.

**Lemma C.5.** *Assuming that OOD calibration error is always greater than the ID calibration error and noting that the range of $h(\cdot)$ is $[0, 1]$, the second term $sd_{\mathcal{H}}(\mathcal{D}_{OOD}, \mathcal{D}_{ID})$ of RHS in Proposition C.3 and C.4 is bound from above by the smallest singular value of the normalized input covariance matrix. That is,*

$$sd_{\mathcal{H}}(\mathcal{D}_{OOD}, \mathcal{D}_{ID}) \leq \frac{d}{\sigma_{min}(\tilde{\Sigma}_{\mathcal{D}_{ID}})}. \quad (20)$$

This can be easily shown by the definition of $sd_{\mathcal{H}}(\mathcal{D}_{\text{OOD}}, \mathcal{D}_{\text{ID}})$,

$$sd_{\mathcal{H}}(\mathcal{D}_{\text{OOD}}, \mathcal{D}_{\text{ID}}) = \sup_{h,h' \in \mathcal{H}} |\mathbb{E}_{\mathcal{D}_{\text{OOD}}}[(h(x) - h'(x))^2] - \mathbb{E}_{\mathcal{D}_{\text{ID}}}[(h(x) - h'(x))^2]| \tag{21}$$

$$\leq \sup_{h,h' \in \mathcal{H}} \frac{\mathbb{E}_{\mathcal{D}_{\text{OOD}}}(h(x) - h'(x))^2}{\mathbb{E}_{\mathcal{D}_{\text{ID}}}(h(x) - h'(x))^2} \tag{22}$$

$$\leq \frac{d}{\sigma_{\min}(\tilde{\Sigma}_{\mathcal{D}_{\text{ID}}})}. \tag{23}$$

Where the first inequality is held by the assumption of the Lemma C.5 and the second inequality is held by Theorem 3 of [11].

Finally, we can derive a more tractable bound for OOD classification and calibration error by plugging the Lemma C.5 into the Proposition C.3 and Proposition C.4. $\qquad\square$

**Discussion on the tightness of proposed bound** Our theoretical analysis was inspired by two previous works that provide OOD generalization error bound [5] and the domain discrepancy bound via the smallest singular value [11]. Intuitively, the inequality of our bounds (ineq.(11)) approaches to equality (becomes tight) under the following conditions:

1. When the outputs of the learned classifier are the same as the outputs of the ideal joint classifier that is trained to minimize both ID and OOD errors [5].

2. Whether the outputs of our classifier or ideal joint classifier are perfectly calibrated [5].

3. The number of training samples $N$ approaches infinite [5].

4. If our classifier is a linear model when the weight vector is the same as the eigenvector corresponding to the smallest eigenvalue of the input covariance matrix [11].

5. When the OOD and ID calibration errors satisfy the relationship: $\varepsilon_{\mathcal{D}_{\text{OOD}}}(h, h^*) = \frac{\varepsilon_{\mathcal{D}_{\text{ID}}}^2(h,h^*)}{(\varepsilon_{\mathcal{D}_{\text{ID}}}(h,h^*)-1)}$.

Although our bound seems quite loose given the high dimensionality of modern machine learning setup, empirical validation in the paper strongly supports the validity of our theory-grounded method.

**Discussion on the gap between reality and assumption** Some ground assumptions derive our bound. First, we assume that ID and OOD have overlapping marginal distributions, whereas the joint distributions do not overlap much. Second, we assume that our hypothesis function has the structure of $h(x) = \sum_{i=1}^{d} h_i(x[i])$. Finally, we assume the pair-wise bi-variate normality of $(x[i], x[j])$ for every $i, j \in [d]$. Given that the input $x$ here is the last layer representation of the visual encoder, ID and OOD have overlapping marginals, but we can not guarantee that the joint distributions of input do not overlap much. For example, there would be a high overlap between ImageNet (ID) and ImageNet-V2 (OOD) in terms of the input joint distributions, but there would not be much overlap in the case of ImageNet-Sketch as an OOD. While it depends on the type of OOD, our evaluation shows robust results across datasets. The second assumption over the structure of our hypothesis function is naturally met if our $h(\cdot)$ is a linear classifier, which is realistic under the modern representation learning paradigm with a heavy feature extractor and a light task-specific head. Lastly, about our last assumption, the representation of neural networks is not usually guaranteed to be pair-wise Gaussian. However, there is rich evidence revealing that the outputs of infinitely wide neural networks (whether they are MLP [31], convolutional neural network [15], or Transformer [62]) are Gaussian processes (GPs) and GPs ensure that every finite collection of their elements are jointly Gaussian. While our numerical analyses are limited to ViT-L scale models, we believe that the current large-scale modeling regime spurs us to explore larger widths [9] where our second assumption becomes more valid [31].

