# OpenReview forum: "Towards Calibrated Robust Fine-Tuning of Vision-Language Models"
_NeurIPS.cc/2024/Conference — NeurIPS 2024 poster_

### Official Review · Reviewer_yVZD · 2024-07-10

**Soundness:** 2
**Presentation:** 2
**Contribution:** 3
**Rating:** 4
**Confidence:** 5

**Summary:**

The paper proposes a novel framework for robust fine-tuning for CLIP. To enhance out-of-distribution accuracy and calibration, the author incorporates a singular-based constraint term, self-distillation, and EMA. Extensive experiments on synthesized data and ImageNet demonstrate that CaRot can achieve better OOD performance and reliable predictions.

**Strengths:**

1. This paper investigates an under-explored but important problem, the calibration of CLIP after fine-tuning.
2. The theoretical analysis between the smallest singular value of image representation and OOD robustness is good.

**Weaknesses:**

1. The overall framework is not well-motivated. Why do we need to incorporate self-distillation and exponential moving average (EMA)? These two techniques are not the main contributions of this paper and are not directly relevant to the primary theoretical analysis of singular value constraints.
2. The connection to Vision-Language Models is weak. The paper does not justify why the proposed soft constraint term is specifically tailored to Vision-Language Models like CLIP. In other words, does this regularization only apply to visual language models?
3. The result analysis is insufficient. The paper does not report the average confidence level, which may lead to the confusion that the calibration improvement may only come from the accuracy improvement.

**Questions:**

1. Can the proposed singular value-based constraint improve other fine-tuning methods, such as FLYP or WISE-FT?

---

> ### Author Rebuttal · Authors · 2024-08-07
>
> **R4-1**
> > The overall framework is not well-motivated. Why do we need to incorporate self-distillation and exponential moving average (EMA)? These two techniques are not the main contributions of this paper and are not directly relevant to the primary theoretical analysis of singular value constraints.
>
> Thanks for your constructive question! As your point, we do not claim those components are our contribution. First, we conduct a novel theoretical analysis on OOD generalization and calibration errors, and the result of these analyses motivates us to pursue better ID calibration as well as increasing the smallest singular value of input covariance matric. Then, we simply employ the self-distillation (SD) with exponential moving average (EMA) as an option among lots of training-time calibration approaches. Our method allows to adopt other alternative calibration approachs, and we provided results on label smoothing (LS) which is a representative train-time calibration regularization in Appendix B (Table D). We further provide results on vanilla knowledge distillation (KD) from zero-shot CLIP as well as varying magnitude of label smoothing for this rebuttal (please refer Fig 3. in the attached PDF). As we see, other type of calibration methods also induce better performance compared with a competative baseline, but the EMA-SD induces best performance. In terms of calibration, EMA-SD can be interpreted as an input dependent label smoothing [14] which adaptively adjusts the smoothed pseudo target label depend on input whereas the label smoothing provides unadaptable fixed pseudo target.
>
> Note that the combination of SD and EMA is also not our originality, and this combination has been widely adopted and validated in the context of vision-language model training [18] as well as self-supervised learning [15,16,17].
>
>
>
> **R4-2**
> > The connection to Vision-Language Models is weak. The paper does not justify why the proposed soft constraint term is specifically tailored to Vision-Language Models like CLIP. In other words, does this regularization only apply to visual language models?
>
> We appreciate the reviewers' keen interest on broader impact of this work. Under the main challenge of robust fine-tuning that is to fine-tune foundation models without loosing its already possessing generalizability to unseen domains, we chose CLIP as our primary focus by following previous works [6,7,8,9]. Since CLIP is widely used for diverse applications not only as a standalone model but as a core component of open-sourced multimodal large language models or text-to-image generative models, we believe that validation on CLIP fine-tuning is crucial in terms of its downstream impact.
>
> However, as the reviwer g5ML and yVZD pointed out, our theoretical results are not confined to CLIP-like VLM, and verifying the applicability of our method to other kind of models will further broaden the impact of our method. Therefore, we expanded our experiment scope to vision-only models, ViT-Base pre-trained by DINOv2 objective [10], using DomainBed benchmark [11], and confirmed the effectives of our proposed framework. Details are elaborated in the global response.
>
> As shown in Table 3 of the PDF file, CaRot shows performance gain on 7 out of 8 cases in terms of Accuracy and ECE on two datsets across two model selection criteria, and achieve relatively smaller performance deviation
>
>
> **R4-3**
> > The result analysis is insufficient. The paper does not report the average confidence level, which may lead to the confusion that the calibration improvement may only come from the accuracy improvement.
>
> To provide a clearer performance improvement, we repeated the experiments three times with different seeds for each method and confirmed that the performance improvement exceeds the error bars. The results are provided in Figure 4 of the attached PDF.
>
>
> **R4-4**
> > Can the proposed singular value-based constraint improve other fine-tuning methods, such as FLYP or WISE-FT?
>
> In Table 3 of our manuscript, we provided the ablation study to validate the effectiveness of our singular value constraint term by plug that constraint to vanilla fine-tuning and FLYP where we can see consistent improvement in terms of OOD performance which aligned with our theoretical analysis. Furthermore, to address the reviewer's concern extensively, we additionally provide the results of the combination of WiSE-FT and our orthogonal constraint term in Table 2 of the attached PDF.

---

> > ### Author Response · Authors · 2024-08-11
> >
> > Dear Reviewer yVZD,
> >
> > We appreciate reviewer yVZD's constructive feedback, which helped us improve our draft further.
> >
> > We have submitted our responses to concerns raised by reviewer yVZD, and we are eager to know if these replies address your concerns!
> >
> > Any further comments or questions are welcome to us.
> >
> > Thank you

---

> ### Comment · Reviewer_yVZD · 2024-08-11
>
> Thanks for the response, and some of my concerns have been addressed.
>
> However, I still have the following two concerns:
>
> 1. The main issue in this paper is the calibration for CLIP. However, the connection between the proposed method and vision-language model (CLIP) is still weak. I think that the proposed method can be applied to most classification models and is not limited to CLIP.
>
> 2. The calibration improvement brought by CaRot seems to come only from accuracy improvement. It does not address the overconfidence issue of CLIP caused by fine-tuning. For example, in Table 2 of the attachment, compared to the baseline, the accuracy of FT increased by 4%, but the ECE only decreased by 3%. Hence, It appears that CaRot is merely a method to improve accuracy rather than a confidence calibration method.
>
> I will keep my original rating.
>
> Best regards,
>
> yVZD

---

> ### Author Response · Authors · 2024-08-12
>
> Dear reviewer yVZD,
>
> We are delighted that some of your concerns have been addressed.
>
> For the remaining concerns,
> * [first concern] As you have pointed out from the original review, we have derived our theory in a common classification setting which is not confined to the vision-language model for the generality of the theoretical statement, and we observe that our OOD error bound-based regularization method is also effective on a vision-only model during the rebuttal.
>   * About the connection between VLM and the proposed method, as described in Section 4.1. of the draft, our orthogonality constraint on the visual projection layer is seamlessly combined with a multimodal contrastive loss so that enables it to be interpreted as a constrained singular value decomposition (SVD) on the cross-covariance matrix of image-text representation pairs.
>   * We appreciate you found that our theory and corresponding method have broad potential applications, and respect your worry. we will reflect your comment to refine the presentation of our revised manuscript. Thanks again.
> * [second concern] We would like to respectfully refute your claim: is CaRot's improved confidence calibration merely due to accuracy improvement?
>   * Allow us to clarify that Table 2 of the attachment is the ablation study of the orthogonality constraint (OC) on FLYP and WiSE-FT, which shows the consistent effect of OOD improvement of OC. **The result of CaRot is not included in that table**. We attached the Table with the results of CaRot below, and demonstrate that **CaRot improves OOD accuracy and ECE as 5.05 (8.7%) and 0.1395 (63.8%). Thus the improvement in terms of ECE is far more significant than the improvement in accuracy.**
>
> | Method     | WiSE-FT | ID Acc | ID ECE | OOD Acc | OOD ECE |
> |------------|---------|--------|--------|---------|---------|
> | FT         | X       | 81.53  | 0.0884 | 57.50   | 0.2186  |
> | FT w/ OC   | X       | 81.45  | 0.0826 | 59.10   | 0.2051  |
> | FLYP       | X       | 82.69  | 0.0635 | 59.46   | 0.1831  |
> | FLYP w/ OC | X       | 82.51  | 0.0651 | 59.51   | 0.1803  |
> | FT         | O       | 82.16  | 0.0820 | 61.22   | 0.1920  |
> | FT w/ OC   | O       | 82.03  | 0.0770 | 61.97   | 0.1829  |
> | FLYP       | O       | 82.98  | 0.0798 | 61.27   | 0.1788  |
> | FLYP w/ OC | O       | 82.80  | 0.0627 | 61.41   | 0.1682  |
> | **CaRot**      | X       | 83.13  | 0.0470 | 62.55   | 0.0791  |
>
> * [second concern] Meanwhile, we partly agree with your statement that improvement in accuracy could contribute to calibration (or vice versa) somewhat. **However, even though the accuracy and calibration are correlated sometimes, they are not a causality.** For example, Figure 2 of Guo et al. 2017 shows that improved accuracy hurts calibration, and Table 1 of Levine et al. presents that CLIP ViT-H-14 and CLIP ViT-B-16 have worse calibration than CLIP ViT-L-14 and CLIP ViT-B-32, respectively, even though they achieve far better classification accuracy. Figure 9 of Dehghani et al. also implies that increased accuracy does not result in improvement in calibration.
>   * We observe the same evidence in Figure 6 of the attached PDF file under our experimental setup. For instance, from the left-most point of FT to the left-most point of FLYP, Accuracy is roughly improved from 81.0 to 82.0 (+1.2%), but the ECE is increased (`worsen`) from 0.06 to 0.08 (+33.3%). Meanwhile, CaRot achieves Accuracy and ECE of 83.0 (+2.4%) and 0.05 (-20%; `become better`). This indicates that **CaRot is not just a method for accuracy improvement, but it is a method for accuracy and calibration simultaneously in a single theory-motivated framework.**
>
> ### Reference
>
> 1. Guo et al. 2017, On Calibration of Modern Neural Networks
> 2. Levine et al. 2023, ENABLING CALIBRATION IN THE ZERO-SHOT INFER- ENCE OF LARGE VISION-LANGUAGE MODELS
> 3. Dehghani et al. 2023, Scaling Vision Transformers to 22 Billion Parameters
>
> We thank you again for the reviewer yVZD's intensive commitment to reviewing our paper and we appreciate valuable comments that contribute to improving the quality of our draft.
>
> Best regard

---

> ### Author Response · Authors · 2024-08-14
> **Sincerely looking forward to your feedbacks**
>
> Dear Reviewer yVZD,
>
> We would like to express our huge gratitude for your invaluable comments so far.
> Your comment has definitely improved the quality of our manuscript and led us to refine our statement further.
> For now, we are wondering if our responses address your remaining concerns. Could you check our responses by any chance?
>
> * We clarify that our theory is not confined to VLM for its generality and broader impact, but our method—integration of multimodal contrastive learning with orthogonality constraint—enables us to interpret our method as a constrained singular value decomposition on the cross-covariance matrix of image-text representation pairs (Please refer to Section 4.1 of our manuscript) during fine-tuning of VLM.
> * Regarding your second concern, we provide counter-examples showing that improved accuracy does not translate to better calibration (with some references) and clarify the experiment results to fix misunderstanding, thereby demonstrating that our CaRot is not just an accuracy-improving method but also promotes non-trivial confidence calibration.
>
> If these replies address your concerns, could we politely ask you to reconsider your rating on our paper?
>
> Sincerely,
>
> The Authors

---

### Official Review · Reviewer_a8to · 2024-07-10

**Soundness:** 3
**Presentation:** 3
**Contribution:** 3
**Rating:** 6
**Confidence:** 3

**Summary:**

his paper aims to improve the accuracy and reduce the calibration error on OOD data for fine-tuning VLM models. The authors first demonstrate that the OOD calibration error and the OOD classification error can be bounded by the ID calibration error and the smallest singular value of the ID input covariance matrix. To address this, the authors apply orthogonal regularization to increase the smallest singular value and use self-distillation to improve the ID calibration. Several experiments on distribution-shifted datasets validate the effectiveness of the proposed method.

**Strengths:**

The paper is well-organized and clearly presented. The theoretical finding that OOD accuracy and calibration error can be bounded by ID error and the smallest singular value is both novel and insightful. The proposed fine-tuning method is simple and intuitive. Numerical analysis of the error bounds and empirical validation on ImageNet OOD benchmarks make the proposed method convincing.

**Weaknesses:**

(1) Some of the notations are misleading. In line 86, the subscript of $x$ denotes the  $n$-th sample in $\mathcal{D} $. However, in line 113, the subscript of $x$ denotes the value of the $i$-th dimension of $x$.

(2) Some of the assumptions are not realistic. In line 113, the authors suppose the hypothesis $h_i$ is a one-dimensional function, each applying to one dimension of the input, which does not hold most of the time. Besides, the authors assume each dimension pair is Gaussian. I am unsure whether the conclusion of Theorem 3.1 still holds if these two assumptions are violated.

(3) The orthogonal constraint is too strong. The finding of Theorem 3.1 states that lifting the smallest singular value is enough to reduce calibration and classification errors. However, the authors implement an orthogonal constraint in the learning procedure, enforcing the features to be an orthogonal basis, which leads to a smaller largest singular value. In Table 3, we can see that applying such regularization alone reduces ID accuracy. I wonder whether applying a regularization that only increases the smallest singular value could achieve higher performance.

**Questions:**

Please refer to the weaknesses.

**Limitations:**

Limitations have been discussed and no negative societal impact has been identified.

---

> ### Author Rebuttal · Authors · 2024-08-07
>
> **R3-1**
>
> Thank you for pointing out the misleading notations. We will revise the notation in the theorem (line 113) as $x[i], \; i=1,...,d$.
>
>
> **R3-2**
>
> We appreciate reviewer's interest and details look on our theorem! As we noted at line 112 of our manuscript, we set the input $x$ as a representation vector produced by the projection layer of CLIP image encoder without lose of generality. Then, our hypothesis function $h(\cdot)$ be a single linear layer classification head that maps the representation vector $x$ to the one-dimensional logit range over $[0,1]$. Here, if we drop the bias term at this classification head, the weight vector $[a_{1},...,a_{d}]$ of $h(\cdot)$ conducts linear combination over $x$, i.e., $h(x)=\sum_{i=1}^{d} h_{i}(x_{i})=a_{1}x_{1}+...+a_{d}x_{d}$, and our first assumption always holds.
>
> About the second assumption of pairwise Gaussian property, as the reviewer a8to's concerns, the representation of modern neural networks are not necessary joint Gaussian for every $(x_{i},x_{j})$. However, our empirical validation with 111 shallow and narrow multi-layer perceptron (MLP) networks in Section 5.1. (See Fig 3.) presents strong evidence that support the validaity of our bounds, and the results of experiments on ImageNet with more wider and deeper network, i.e., CLIP ViT-B, are also well-aligned with our theory.
>
> Moreover, there is a bunch of works revealing that the outputs of infinitly wide neural networks (whether they are MLP [19], convolutional neural network [20], Transformer [21]) are Gaussian processes (GPs) and GPs ensure that every finite collection of their elements are jointly Gaussian. Therefore, our second assumption becomes valid as the network's layer-wise width being increased. While our numerical analyses are limited to ViT-L scale models, we believe that current large-scale modeling regime spurs us to explore larger width [22] where our second assumption becomes more valid [19]. Meanwhile, we recognized some theoretical results on the Gaussianity of neural network representations under finite-width regime [23], and we plan to explore the relaxation of our assumption based on those insights. Thanks again for the constructive criticism.
>
> **R3-3**
>
> As the reviewer pointed out, there are other possible design choices for the smallest singular value regularization. One straightforward way is to directly regularize only the smallest singular value. In Table 1 of attached PDF, we provide the result of direct singular value regularization method. This approaches achieve smaller ID-OOD performance gap compared with our current method, which is aligned with our theoretical results (see appendix) and that of [5]. However, direct singular value regularization method entails singular value decomposition (SVD) on the covariance matrix size of $\mathbb{R}^{d \times d}$, which require cubic time complexity over the feature dimension which is significantly increase computational cost especially on the large-scale modeling regime. To this end, we employ the orthogonality regularization on the final projection matrix $W$ of visual encoder which indirectly (yet effectively) increase the smallest singular value of input representation and its covariance matrix.
>
> To be specific, enforcing the projection matrix to be orthogonal matrix $O$ ensures that the rank of input representation and its covariance matrix produced by our method is closer to the upper bound rank than an y other possible projection matrices $W$ without the constraint as below,
>
> \begin{equation}
> \begin{split}
> \text{rank}(\tilde{Z}^{T}\cdot\tilde{Z}) &= \text{rank}(\tilde{Z}) \\\\
>  &= \text{rank}(W\cdot Z^{T}) \\\\
>  &\le \text{rank}(O\cdot Z^{T}) \\\\
>  &= \text{rank}(\hat{Z}) \\\\
>  &= \text{rank}(\hat{Z}^{T}\cdot\hat{Z})
> \end{split}
> \end{equation}
>
> where $Z$ is input feature before projection, and $\tilde{Z}$ and $\hat{Z}$ denote input representations obtained by projection matrix $W$ and $O$, respectively. However, the SVD has been continuously studied to improve computational efficiency due to its popularity, and we found some approximation-based fast SVD methods such as `scipy.sparse.linalg.svds` though it is non-differentiable. Devising the fast differentiable SVD-based new fine-tuning method, which is well-aligned with our theory, would be very exciting future work direction and we appreciate the reviewer a8to's valuable query.
>
> Meanwhile, the reduced ID accuracy by applying orthogonal constraint can be interpreted as compensation for improved OOD generalizability, which is the same in the case of direct SVD-based regularization. Intuitively, this phenomenon indicates that a large smallest singular value encourages the model to capture more diverse features while compromising ID-specific discriminative biases somewhat. We agree that our ultimate goal should be achieve good OOD generalization without compromising ID adaptation capability. We leave devising a method that produces better ID-OOD trade-off as our future work.

---

> > ### Author Response · Authors · 2024-08-11
> > **Reminder for discussion, Reviewer a8to**
> >
> > Dear Reviewer a8to,
> >
> > We appreciate reviewer a8to's valuable comments that significantly contribute to improving our manuscript.
> >
> > We have submitted our responses to reviewer a8to's concerns, and we want to know if these replies address your concerns!
> >
> > Any further comments or questions are welcome to us.
> >
> > Thank you

---

> > > ### Comment · Reviewer_a8to · 2024-08-11
> > >
> > > Dear Authors,
> > >
> > > Thank you for your detailed responses, which have addressed most of my concerns. I would like to kindly request that you revise the notations, add an explanation of the assumption of the one-dimensional function, and discuss other possible design choices for the smallest singular value regularization in the updated version.
> > >
> > > Before I raise my rating, I have one additional question regarding the ablation study in Table 4 and Table 3: In Table 3, applying orthogonal regularization seems to reduce ID accuracy. However, in Table 4,$\lambda_{OC} = 0.1$ results in higher ID performance than $\lambda_{OC} = 0$, which appears to contradict the conclusion of Table 3. Could you please further explain this discrepancy?
> > >
> > > Best regards,

---

> ### Author Response · Authors · 2024-08-12
>
> Dear reviewer a8to,
>
> 1. We truly appreciate your suggestion on fixing notation, clarifying assumptions of the theory, and discussing potential alternative design choices of constraint terms. We promise to update our manuscript thoroughly, as you pointed out, and this will significantly refine the presentation quality from now on.
> 2. We also appreciate your further interest in the behavior of orthogonality constraint (OC) and the consideration of raising the score!
>
>
> Allow us to clarify the experimental setup for Table 3 and Table 4 of our manuscript. The left side of **Table 4 is the results of the ablation study where the self-distillation (SD) regularization term is applied with a 1.5 multiplier**. That is, the first and third rows of Table 4 are equal to the seventh and eighth rows of Table 4. For a clear comparison, we insert a table below that includes results with and without SD under varying OC magnitudes. Meanwhile, As you can see **by comparing the first and second, and the fifth and sixth rows of Table 3, orthogonality constraint (`without SD term`) slightly tradeoffs ID accuracy**. This phenomenon is also observed through additional ablation studies with the WiSE-FT method in Table 3 of the rebuttal attachment PDF.
>
> | Objective | OC  | ID Acc | ID ECE | OOD Acc | OOD ECE |
> |-----------|-----|--------|--------|---------|---------|
> | MCL       | 0   | 82.69  | 0.0635 | 59.40   | 0.1836  |
> | MCL       | 0.1 | 82.48  | 0.0652 | 59.41   | 0.1807  |
> | MCL       | 0.2 | 82.51  | 0.0651 | 59.51   | 0.1803  |
> | MCL w/ SD | 0   | 83.03  | 0.0523 | 62.28   | 0.0772  |
> | MCL w/ SD | 0.1 | 83.18  | 0.0511 | 62.42   | 0.0779  |
> | MCL w/ SD | 0.2 | 83.13  | 0.0470 | 62.55   | 0.0791  |
>
>
> **Therefore, we derive non-contradictive conclusions from Tables 3 and 4 of the manuscript, indicating that orthogonality constraint alone somewhat compromises ID adaptation capability for OOD generalization, while this tradeoff is mitigated when the SD is applied together, which is our final learning objective.**
>
> We speculate that,
> 1) When the orthogonality constraint is applied alone, the model is enforced to capture diverse features for OOD generalization yet without any restriction on the type and priority of learned features. While this contributes to enhancing OOD generalization, **diverse features without prioritization might compromise strong ID performance.**
> 2) However, the SD regularization produces input-dependent soft labels that hold similarity structures between classes. This allows the model to learn diverse features while **putting a higher priority on features shared across similar classes** (judged by the EMA teacher model) so that the features are beneficial not only for OOD generalization but also for ID adaptation. We can understand the joint use of the orthogonality constraint and self-distillation regularization induces a narrower solution set, which potentially induces better generalization (Huh et al. 2024).
>
> [Huh et al. 2024] The Platonic Representation Hypothesis
>
> Thanks again for your constructive comment so far.
>
> Best regard

---

> > ### Comment · Reviewer_a8to · 2024-08-13
> >
> > The authors have addressed my concerns effectively, and I have increased my score accordingly.

---

### Official Review · Reviewer_nZD2 · 2024-07-12

**Soundness:** 3
**Presentation:** 3
**Contribution:** 3
**Rating:** 6
**Confidence:** 3

**Summary:**

VLMs have shown to be effective in a wide-area of applications, though they can fail under certain domain shifts. In this work, the authors first observe that the the upper bound of generalization and calibration under domain shifts is bounded by the ID calibration error and the smallest singular value of the ID covariance matrix. Building upon this intuition, the authors then propose a novel fine-tuning scheme for VLMs in the shape of a contrastive objective and a self-distillation technique. Experimental evaluation on Imagenet and quite a few of its variants are presented to show the effectiveness of the proposed approach.

**Strengths:**

- The motivation for the work is neatly presented with a clear link between the motivation and the proposed method. While the exact method does not depend on Theorem 3, it is still a good way to explain why the proposed method can serve as a proxy to reduce OoD unreliability of the models. Sections 3 and 5.1 are really helpful in this regard.
- Throughout the work, the design choices are explained clearly in a technically sound manner with intuitive connections being made between the utilized concepts and the math behind.
- The experimental benchmarks chosen for evaluation seems to be adequate as it involves a rather wide range of Imagenet variants, from those that are much more similar to Imagenet like ImagenetV2 to Imagenet-S. From the results of these experiments, it can be seen that the proposed method mostly brings significant improvements for the domain-shift settings.

**Weaknesses:**

- In certain cases, such as Imagenet-R on Table 2 and 3, it appears that the zero-shot CLIP is better than _any_ of the fine-tuning methods including the proposed approach in terms of _both_ the accuracy and calibration. Furthermore, the calibration after fine-tuning with CaRoT is worse than the zero-shot CLIP under harsher domain-shift benchmarks, namely the ObjectNet, Imagenet-{A, S, R}. I wonder how would the Table 2 look like had the model had access to multiple training environments (e.g maybe having a subset of Imagenet-S during fine-tuning, then evaluated on Imagenet-R) as it is often the case for domain generalization benchmarks [A].
- On Tables 2 and 3, the proposed approach seems to be falling behind of the other methods in terms of Imagenet accuracy, which may limit its usefulness for a wider range of applications.
- It would have been good to have Imagenet-C [B] here as well, as it is perhaps the most commonly used among these variants for domain shift benchmarking. In particular, having some analyses based different types of corruptions and under different severity levels could have provided more insights into the limitations and strengths of the proposed approach.
- One of the minor issues I can see with the work is regarding the usage of the term "OOD", especially since the benchmarks used vary significantly from Imagenet-V2 (which was designed to be distributionally as similar as possible to the original Imagenet) to Imagenet-S. While I acknowledge that the authors clarify what they mean under Section 2, I still encourage them to check out [C] for a nice template on works involving the term.



[A] Guljarani et al., "In search of lost domain generalization", ICLR 2021

[B] Hendrycks et al., “ Benchmarking Neural Network Robustness to Common Corruptions and Perturbations”, ICLR 2019

[C] Farquhar et al., "What ‘Out-of-distribution’ Is and Is Not", NeurIPS-W 2022

**Questions:**

I wonder what the authors think about the first weakness I have described above and it would be great to see detailed Imagenet-C results.

**Limitations:**

The limitations are somewhat discussed in the conclusion. I appreciate the fact that they have stated that they could not include larger models due to computational constraints which prompted me to avoid asking for how the proposed method would perform with larger models trained with much larger data.

---

> ### Author Rebuttal · Authors · 2024-08-07
>
> **R2-1**
> > (a) In certain cases, such as Imagenet-R on Table 2 and 3, it appears that the zero-shot CLIP is better than any of the fine-tuning methods including the proposed approach in terms of both the accuracy and calibration. Furthermore, the calibration after fine-tuning with CaRoT is worse than the zero-shot CLIP under harsher domain-shift benchmarks, namely the ObjectNet, Imagenet-{A, S, R}.
>
> As the reviewer pointed out, zero-shot CLIP shows strong performance in both terms of accuracy and calibration error. The robust fine-tuning literature focused on preserving (or improving) the OOD accuracy after fine-tuning CLIP on ID data. In this work, we expanded the metrics of interest to OOD calibration error and observed that existing baselines sacrifice the confidence calibration as well as OOD generalization errors. As shown in Table 1 and 2 in the manuscript, compared with other fine-tuning methods, CaRot achieves the best accuracy in all OOD datasets and minimum calibration error in four out of five OOD datasets.
>
> We agree that improving OOD generalization and calibration error beyond zero-shot CLIP should be the ultimate goal of robust fine-tuning research, and we leave this as our future work.
>
> > (b) I wonder how would the Table 2 look like had the model had access to multiple training environments (e.g maybe having a subset of Imagenet-S during fine-tuning, then evaluated on Imagenet-R) as it is often the case for domain generalization benchmarks [A].
>
> We appreciate the reviewer for suggesting such a meaningful evaluation setup to investigate the versatility of CaRot. As recommended, we conducted experiments on multi-source domain generalization with DomainBed [11]. We used PACS and VLCS datasets, where each dataset consists of four domains of the same class labels (i.e., covariate shift). Following the leave-one-out setting, we train the model on three domains and test on the unseen remaining domain (please refer to global response for details).
>
> The results in Table 3 of the attached PDF show that CaRot on top of ERM++ achieves the best performance on three out of four cases in terms of Accuracy and all four cases in terms of ECE. Especially, on PACS dataset of training-domain validation model selection setup, CaRot improves the accuracy of ERM++ from 95.0 to 96.2 and ECE of ERM++ from 0.025 to 0.013 that are signifcant given that the absolute performance of ERM++ is already very competative. This indicates that CaRot can be effectively adopted in setups where multiple training domains are available, enhancing both the accuracy and calibration of the algorithm.
>
>
> **R2-2**
>
> We acknowledge the reviewer's concern regarding the marginal underperformance of CaRot on ID (ImageNet) data compared to Lipsum-FT. However, the performance gain on the OOD data (61.04 -> 62.55; **2.5%**), which is significantly larger than the loss of ID performance (83.30 -> 83.13; **0.2%**), implies better effective robustness of CaRot. This robustness is crucial for many safety-critical real-world applications, where the ability to generalize to unseen data is often more important than slight improvements on the training data.
>
> **R2-3**
>
> We appreciate the reviewer's suggestion, and we evaluated CaRot on ImageNet-C [13] consisting of 15 synthetic corruptions with five severities. We report the results averaged over severities in Figure 1 of the attached PDF. The results show that CaRot consistently outperforms other fine-tuning methods in terms of Accuracy and ECE across all corruption types. Specifically, for the coarser corruption such as Snow, Forst, Fog, Bightness and Contrast that are more natural type of shifts compared to others, CaRot greatly ourperforms baseline methods whereas on the finer corruption such as Elastic transform the performance gain by CaRot relatively diminishes. Due to the space constraints, we attached box plots only for brightness and elastic transform in Figure 2 of the PDF.
>
> **R2-4**
>
> We appreciate the reviewers suggestion on clarifying the terminology of "OOD". We will clarify the term in the manuscript regarding the recommended reference.

---

> > ### Comment · Reviewer_nZD2 · 2024-08-10
> >
> > I appreciate the hard work the authors have put into the rebuttal. They seem to have answered my points fairly well, especially with the detailed DomainBed and Imagenet-C experiments and visuals. I am increasing my score accordingly.

---

> ### Author Response · Authors · 2024-08-11
>
> We are so delighted that our rebuttal addresses your concerns and questions!
> Thank you for your valuable comments and for taking the time to review our paper thoroughly.

---

### Official Review · Reviewer_g5ML · 2024-07-15

**Soundness:** 3
**Presentation:** 3
**Contribution:** 3
**Rating:** 5
**Confidence:** 2

**Summary:**

This research paper presents a novel fine-tuning approach for improving out-of-distribution (OOD) generalization and calibration in Vision Language Models (VLMs). By identifying a shared upper bound for OOD accuracy and calibration errors, the authors develop a constrained multimodal contrastive loss framework enhanced by self-distillation. Empirical validation and tests on ImageNet benchmarks demonstrate the method's effectiveness in enhancing both OOD accuracy and calibration error.

**Strengths:**

- The proposed approach is well motivated by theory. The connection of out-of-distribution (OOD) generalization and calibration in terms of the smallest singular value of the input covariance matrix of in-distribution (ID) data is interesting and novel to my knowledge.
- The experiments are extensive and the results are good on both fronts.

**Weaknesses:**

- The theory part is a bit obscure. Is there an intuitive interpretation of the smallest singular value of ID input covariance matrix in the studied context? How tight is the upper bounds (Theorem 3.1)?
- Were the main experiments repeated multiple times? If so, can the authors provide the error bars for the main experiments?  - The paper limits the scope to VLM fine-tuning, but does it also apply to fine-tuning other kinds of models?

**Questions:**

Please see the weaknesses.

**Limitations:**

The authors discussed one limitation of the work. I don't notice any potential negative societal impact.

---

> ### Author Rebuttal · Authors · 2024-08-07
>
> ## R1 (g5ML)
> **R1-1**
> > (a) The theory part is a bit obscure. Is there an intuitive interpretation of the smallest singular value of ID input covariance matrix in the studied context?
>
> We derive the smallest singular value into the bounds for OOD errors through two steps: 1) We frist derive OOD calibration and generalization error bounds with ID calibration error and H-square disagreement between ID and OOD, 2) we then introduce the reciprocal of the smallest singular value of ID covariance matrix as an upper bound of H-square disagreement. By substituting H-square disagreement (which requires access to OOD data) with the smallest singular value term, we now obtain OOD error bounds that depend soley on ID data.
>
> According to Theorem 3.1, the upper bound on the OOD classification and calibration errors decreases as the smallest singular value increases. Intuitively, enforcing larger smallest singular values can be interpreted as inducing a larger effective rank of the input representation and its covariance matrix. The effective rank measures how evenly the singular values are distributed, indicating that the model learns more diverse features [1] and is less likely to focus on ID-specific biased attributes, thus improving OOD generalizability. This concept of de-correlating the covariance of input representations to increase the effective rank is well-studied in the context of self-supervised learning, which aims to learn transferable representations [2,3].
>
> Motivated by this theoretical insight, we developed a method that encourages a high effective rank of input representations and their covariance matrix by applying a soft constraint to ensure the last projection layer, $W$, functions as an orthogonal projection matrix, $O$. This ensures that the rank of ID input representation and its covariance matrix produced by our method is closer to the upper bound rank than any other possible projection matrices, $W$, without this constraint as below,
>
> $$
> \begin{split}
> \text{rank}(\tilde{Z}^{T}\cdot\tilde{Z}) &= \text{rank}(\tilde{Z}) \\\\
>  &= \text{rank}(W\cdot Z^{T}) \\\\
>  &\le \text{rank}(O\cdot Z^{T}) \\\\
>  &= \text{rank}(\hat{Z}) \\\\
>  &= \text{rank}(\hat{Z}^{T}\cdot\hat{Z})
> \end{split}
> $$
>
> where $Z$ is input feature before projection, and $\tilde{Z}$ and $\hat{Z}$ denote input representations obtained by projection matrix $W$ and $O$, respectively.
>
> > (b) How tight is the upper bounds (Theorem 3.1)?
>
> Our theoretical analysis is inspired by two previous works that provide OOD generalization error bound [4] and the domain descrepancy bound via the smallest singular value [5].
>
> Intuitively, the inequality of our bounds (1) in the manuscript approaches to equality (becomes tight) under the following conditions:
> * i) When the outputs of learned classifier are same with the outputs of ideal joint classifier that is trained to minimize the both ID and OOD errors [4].
> * ii) Whether the outputs of our classifier or ideal joint classifier are perfectly calibrate [4].
> * iii) The number of training samples $N$ approaches infinite [4].
> * iv) If our classifier is a linear model, when the weight vector is same with the eigen vector corresponding to the smallest eigen value of input covariance matrix [5].
> * v) When the OOD and ID calibration errors satisfy the relationship: $\varepsilon_{OOD}(h,h^{\*})=\varepsilon^{2}_{ID}(h,h^{\*})$ / $(\varepsilon _{ID}(h,h^{*}) - 1)$
>
>
> While we leave the exact mathematical statement on the tightness of our bounds to future work, we demonstrate the validity of our bounds as shown in Section 5.1. This is further supported by promising results on real datasets presented in Section 5.2. Additionally, we provide empirical validation of the tightness of our OOD calibration error bound in Figure 5 of PDF.
>
> We compute the right-hand side of our bound as $\varepsilon_{\text{ID}}(h)$+ $\\frac{1}{\\sigma_{min}(\tilde{\Sigma_{ID}})}$ +$\\Delta$  by neglecting the $\mathcal{O}(\cdot)$ term, assuming we have sufficiently large training samples, and we approximate the joint optimal risk $\Delta$ as a sum of ID and OOD errors from the models trained on ID and OOD train set, simultaneously. We see that the estimated upperbound is placed between the $\varepsilon_{{OOD}}(h) + 2\text{std}(\varepsilon_{OOD}(h))$ and $\varepsilon_{{OOD}}(h) + 3\text{std}(\varepsilon_{{OOD}}(h))$ region in average, which is not significantly deviated from true OOD error.
>
> **R1-2**
> > Were the main experiments repeated multiple times? If so, can the authors provide the error bars for the main experiments?
>
> We repeated the experiments three times with different random seeds for each method and confirmed that the performance improvement exceeds the error bars. The results are provided in Figure 4 of the attached PDF. We will also update the tables in the manuscript accordingly.
>
> **R1-3**
> > The paper limits the scope to VLM fine-tuning, but does it also apply to fine-tuning other kinds of models?
>
> Under the main challenge of fine-tuning foundation models without losing their inherent generalizability to unseen domains, we chose CLIP as our primary focus by following previous works [6,7,8,9].
>
> However, as the reviwer g5ML pointed out, our theoretical results are not confined to vision-language models (VLMs). Verifying the applicability of our method to other kind of models will further broaden the impact of our method. Therefore, we expanded our experimental scope to vision-only models, specifically DINOv2 [10] pre-trained ViT-base, using DomainBed benchmark [11], and confirmed the effectives of our proposed framework. The results are provided in Table 3 of the attached PDF.
>
> We see that while ERM++ method already achieve superior performance compared with other baselines, applying CaRot objective to ERM++ further improves its performance in terms of Accuracy and ECE, and shows the best results in two considered datasets.

---

> > ### Comment · Reviewer_g5ML · 2024-08-08
> >
> > Thank you for the further clarifications.
> >
> > I get the idea that a higher effective rank of the feature covariance matrix implies higher feature diversity. However, I'm still confused about the role of minimizing $\lVert W_v W_v^T - I\rVert^2_F$ in achieving such a goal.
> >
> > Suppose $W_v = I$, which is the best we can get. This means the projection layer preserves the effective rank $r$ of the feature from the previous layer, so we have $r' = r$ where $r'$ is the effective rank of the projected feature. The question is, does this have any effect on $r$ itself? To me, it seems the answer is no because the effective rank $r$ of the pre-projection feature may be low even if $\lVert W_v W_v^T - I\rVert^2_F = 0$, i.e., they are independent.
> >
> > Empirically, the ablation study (Table 3) suggests that the proposed constraint, i.e., $\lVert W_v W_v^T - I\rVert^2_F = 0$, plays a small role. The main improvement is brought by FLYP and SD but they are not the main contribution of this work (as partly mentioned by Reviewer yVZD). Could the authors please further comment on this?

---

> ### Author Response · Authors · 2024-08-08
>
> Thank you for your active query!
>
> ### 1. Rank comparison
>
> > Allow us to recap the rank of matrix multiplication,
> $\\text{rank}(AB) \\le \text{min}(\\text{rank}(A),\\text{rank}(B))$
> where the equality only holds whether matrix $A$ or $B$ has full rank.
>
> Then, for the neural network representation of our case, **the post-projection feature always has a smaller rank compared to the pre-projection feature if the projection matrix $W$ is not full rank**. There is a solid theoretical background about this so-called **rank diminishing phenomenon** [Feng et. al 2022].
>
> As you said, our constraint can't increase the rank of the pre-projection feature and just preserve it in the ideal case; however, this preservation induces the higher rank of the post-projection feature compared with the baseline, which does not encourage the projection matrix to be full-rank.
> Therefore, our method induces a higher effective rank of the feature covariance matrix compared to baseline methods, which do not enforce rank preservation, and our method induces learning of diverse features that contribute to better OOD generalization.
>
> _We really appreciate this valuable query, and will revise our manuscript to clarify further the connection between singular values, rank, and OOD generalization._
>
> ### 2. Significance of performance gain
> We would like to emphasize that, in Table 2 of the attached PDF file, our orthogonal constraint, OC for short, **consistently improves the OOD accuracy and ECE in all 8 cases**, as well as a case of the toy experiment in the left side of Figure 3 in the manuscript. While the amount of improvement might be seen as small, we believe **this consistent improvement (9 out of 9) indicates strong evidence of the effectiveness of our theory-motivated rank-preserving constraint for OOD generalization**, which is complementary to the benefits from the FLYP or SD. Moreover, the amount of performance gain is sometimes significant, e.g., +1.6 in terms of accuracy of FT w/ and w/o OC, -0.01 in terms of ECE of FLYP-WiSE w/ and w/o OC.
>
> Please do not hesitate to ask us any further questions; we will be delighted to have an extended discussion with the reviewer g5ML.
>
> ---
>
> ### reference
> [Feng et al. 2022] Rank Diminishing in Deep Neural Networks

---

> > ### Comment · Reviewer_g5ML · 2024-08-10
> >
> > Thank you for addressing my queries. I have no further questions or comments. I would like to keep my score for now.

---

> > > ### Author Response · Authors · 2024-08-11
> > > **Further Suggestions for Improvement**
> > >
> > > Thank you for your detailed feedback and inquiry so far!
> > >
> > > Your comments were highly valuable to us. We have thoroughly addressed the issues raised by Reviewer g5ML.
> > > We believe these revisions will enhance our manuscript significantly.
> > >
> > > **We respectfully inquire if you could consider raising the score, given that all of your concerns are addressed.
> > > If not, we are interested in understanding if there are any additional concerns or suggestions that we could address to improve the submission further.**
> > >
> > > Again, we would like to express our gratitude for your commitment to reviewing our paper thoroughly and providing constructive feedback, which helps us refine our work.

---

> > > > ### Comment · Reviewer_g5ML · 2024-08-11
> > > >
> > > > My initial concerns are addressed and I really appreciate the authors' detailed, thoughtful responses. They helped me better understand the paper and see clearer its merits and defects.
> > > >
> > > > The reason I would like to keep my score is that the improvement brought by the proposed orthogonality constraint appears to be somewhat marginal. The constraint only deals with the loss of feature diversity at the last projection layer, disregarding potential losses at earlier stages, which are probably more consequential. For future improvement, I wonder if a similar constraint can be directly applied to the features themselves (as the theory implies) rather than the layer weights.
> > > >
> > > > Finally, the overall method, CaRot, feels like a naive combination of three separate components. In particular, as pointed out by Reviewer yVZD, the connection of the method to vision-language models is weak. Although the authors have presented further results on vision-only models, this does not fix the framing of the paper (fixing it would probably require a major revision).
> > > >
> > > > In all, while I am inclined to accept the paper, the impact of the paper in its current form may be limited so I think a borderline accept is adequate.

---

> > > > > ### Author Response · Authors · 2024-08-12
> > > > >
> > > > > Dear Reviewer g5ML,
> > > > >
> > > > > We sincerely appreciate your further clarification, and we want to express our huge gratitude for the positive rating incline to acceptance!
> > > > >
> > > > > In this work,
> > > > > * we **provide a novel theoretical bound on both OOD generalization and calibration errors in a single unified formulation that solely requires ID-related terms**, i.e., ID calibration error and the smallest singular value of the ID covariance matrix.
> > > > > * **CaRot, which is designed to minimize this error upper bound, was extensively validated on the large-scale dataset across diverse backbones**.
> > > > > * Through the ablation studies, we confirm that the **adoptions of each component**, i.e., orthogonality constraint and calibration regularization, of CaRot **induce consistent improvements in the OOD generalization and calibration**.
> > > > >
> > > > > We respect your viewpoint on the effectiveness of the orthogonality term and thank you for the opinion of direct regularization on the feature singular value. While our orthogonality constraint was introduced to promote a larger smallest singular value of the feature covariance matrix, **it also allows us to view our multimodal contrastive loss as a constrained singular value decomposition (SVD) on the cross-covariance matrix of image-text representation pairs that helps interpretation of our method**.
> > > > >
> > > > > In Table 1. of the attached PDF, we included a result of the direct feature regularization method based on the singular value decomposition (SVD) and using the negative smallest singular value as an additional loss term. This method achieved promising outcomes in terms of OOD performance. We will further investigate the other direct constraint terms in future work.
> > > > >
> > > > > About your second concern 'weak linkage between the method and VLM, framing of paper', while the combination of each component has originality in terms of theoretical motivation and interpretation, we respect and will accommodate your opionion to increase the impact of our work.
> > > > >
> > > > > Thanks again for your invaluable feedback so far. We believe that our manuscript will be significantly improved.
> > > > >
> > > > > Best regard

---

### Author Rebuttal · Authors · 2024-08-07

## Summary of Rebuttal

*We sincerely thank all four reviewers for their constructive feedback and valuable comments.*

**The strengths of our work, as highlighted by reviews:**
* The motivation behind this work is clear, addressing an important but under-explored problem.
* There is a strong connection between theoretical findings and the proposed method, which is simple and intuitive.
* The experimental setting is well-chosen, and the extensive results, coupled with theoretical analysis, demonstrate the method's effectiveness.

**Our responses to the reviews:**
* **[Details on orthogonality constraint]** We provided an intuitive interpretation of our theoretical finding about the connection between the smallest singluar value and out-of-distribution (OOD) generalization. We also justified the implementation of orthogonal constraint and demonstrated its effectiveness through extended ablation studies.
    * R1(g5ML)-1(a), R3(a8to)-3, R4(yVZD)-4
* **[Details on EMA SD]** We explained the design motivation for EMA SD with results of extended ablation studies.
    * R4(g5ML)-1
* **[Generality of method]** We demonstrated the general applicability of our method by expanding the experimental setup to include vision-only models, multi-source domain generalization, and synthetic shift settings.
    * R1(g5ML)-3, R2(nZD2)-1, R2(nZD2)-3, R4(yVZD)-2
* **[Repeated experiments]** We validated our results by repeating the experiments with multiple seeds.
    * R1(g5ML)-2, R4(yVZD)-3
* **[Elaboration on results]** We emphasized our performance improvement and provided interpretations of the results.
    * R2(nZD2)-1, R2(nZD2)-2
* **[Elaboration on theorem]** We justified the assumptions of Theorem 3.1.
    * R1(g5ML)-1(b), R3(a8to)-2
* **[Notation]** We clarified our notations and terms.
    * R2(nZD2)-4, R3(a8to)-1

## DomainBed Setup
> To address reviewers' concerns, we conduct several experiments. Among them, we elaborate the setup for "CaRot for the **vision-only pre-trained model** fine-tuning on the **multi-source (train) domains**", which received the most inqueries.
* We validate the applicability of CaRot to the vision-only pre-trained model, `DINOv2 ViT-base` [10], on two represenative domain generalization benchmarks, `PACS` and `VLCS`. Each dataset consists of four domains with the same class labels (i.e., covariate shift). Following the leave-one-out training setting, we trained the model on three domains and tested it on the unseen remaining domain.
* We consider eight baseline methods [24-31], and we validate the CaRot objective by incorporating the orthogonal constraint term and self-distillation with exponential moving average term into the training pipeline of `ERM++` [31].
* By following DomainBed [11], we set a fixed hyperparameter sweep budget of 10 for all baseline methods. Each hyperparameter configuration was run three times, resulting in $120 = 10\text{(hyperparameter)}  *  3\text{(seed)}  *4  \text{(domain)}$ runs per algorithm on a dataset.
* Among three types of model selection of DomainBed, we report resulst with training domain validation and test-domain validation (oracle) selection strategies.

## Reference
1. Assessing the downstream performance of pretrained self-supervised representations by their rank, Garrido et al. 2023
2. Self-Supervised Learning via Redundancy Reduction, Zbontar et al. 2021
3. Variance-Invariance-Covariance Regularization for Self-Supervised Learning, Bardes et al. 2022
4. A theory of learning from different domains, Ben-David et al. 2010
5. First Steps Toward Understanding the Extrapolation of Nonlinear Models to Unseen Domains, Dong and Ma 2022
6. Robust fine-tuning of zero-shot models, Wortsman et al. 2022
7. Fine-Tuning can Distort Pretrained Features and Underperform Out-of-Distribution, Kumar et al. 2022
8. Improved finetuning of zero-shot vision models, Goyal et al. 2023
9. Robust Fine-Tuning of Zero-Shot Models Using Random Text Guidance, Nam et al. 2024
10. Learning Robust Visual Features without Supervision, Oquab et al. 2023
11. In Search of Lost Domain Generalization, Gulrajani and Lopez-Paz 2020
12. Measuring Robustness to Natural Distribution Shifts in Image Classification, Taori et al. 2020
13. Benchmarking Neural Network Robustness to Common Corruptions and Perturbations, Hendrycks and Dietterich, 2019
14. Self-Distillation as Instance-Specific Label Smoothing, Zhang and Sabuncu 2020
15. Bootstrap Your Own Latent A New Approach to Self-Supervised Learning, Grill et al. 2020
16. A General Framework for Self-supervised Learning in Speech, Vision and Language, Baevskil et al. 2022
17. Multi-Mode Online Knowledge Distillation for Self-Supervised Visual Representation Learning, Song et al. 2023
18. Vision and Language Representation Learning with Momentum Distillation, Li et al. 2021
19. DEEP NEURAL NETWORKS AS GAUSSIAN PROCESSES, Lee et al. 2018
20. DEEP CONVOLUTIONAL NETWORKS AS SHALLOW GAUSSIAN PROCESSES, Garriga-Alonso et al. 2019
21. Wide Feedforward or Recurrent Neural Networks of Any Architecture are Gaussian Processes, Yang 2021
22. Scaling Vision Transformers to 22 Billion Parameters, Dehghani et al. 2023
23. On the infinite-depth limit of finite-width neural networks, Hayou 2023
24. Statistical learning theory, Vapnik 1998
25. Distributionally robust neural networks for group shifts: On the importance of regularization for worst-case generalization, Sagawa et al. 2019
26. Self-supervised Contrastive Regularization for Domain Generalization, Kim et al. 2021
27. Invariance Principle Meets Information Bottleneck for Out-of-Distribution Generalization, Ahuja et al. 2022
28. Optimal Representations for Covariate Shift, Ruan et al. 2022
29. Invariant Causal Mechanisms through Distribution Matching, Chevalley et al. 2022
30. Probable Domain Generalization via Quantile Risk Minimization, Eastwood et al. 2022
31. An Improved Baseline for Domain Generalization, Teterwak et al. 2023

---

### Decision · Program_Chairs · 2024-09-25

**Decision:**

Accept (poster)

**Comment:**

The paper proposes a method for improving out-of-distribution confidence calibration in vision-language models. The proposed method involves finetuning with a constrained multimodal contrastive loss that minimizes an upper bound on out-of-distribution calibration. Reviewers were generally positive about the paper, highlighting the significance of the theoretical result and the excellent presentation of the paper, but also raised concerns about the fact that the proposed method only leads to small improvements in performance and that the proposed regularization term is not specific to vision-language models. While I sympathize with these concerns, I believe that---although the framing of the paper likely reduces the impact of the theoretical results on the broader community---the contributions of the paper are sufficiently significant and of sufficient interest to the NeurIPS community to warrant publication. In conclusion, I recommend acceptance.